# Exposing Mixture and Annotating Confusion for Active Universal Test-Time Adaptation

**Jiayao Tan**[1,2*], **Fan Lyu**[3*], **Chenggong Ni**[2], **Fuyuan Hu**[2†], **Wei Feng**[4†], **Rui Yao**[5]

[1]School of Artificial Intelligence, Tianjin University
[2]Suzhou University of Science and Technology
[3]Computer Vision Centre, Universitat Autònoma Barcelona
[4]School of Computer Science, Tianjin University
[5]China Unviersity of Mining and Technology
`jiayaotan@tju.edu.cn, fanlyu@cvc.uab.cat, wfeng@ieee.org`
`{cgn@post,fuyuanhu@mail}.usts.edu.cn, ruiyao@cumt.edu.cn`

## Abstract

Universal Test-Time Adaptation (UTTA) tackles the challenge of handling both class and domain shifts in unsupervised settings with streaming test data. However, existing UTTA methods are often limited to minor shifts and heavily rely on heuristic rules. To advance UTTA under dual shifts, we propose a novel framework, Active Universal Test-Time Adaptation (AUTTA), and instantiate it with Exposing Mixture and Annotating Confusion (EMAC), which incorporates active human annotation into the UTTA setting. To select appropriate samples for annotation in AUTTA, we first identify the mixed regions of target-domain samples under dual shifts, thereby exposing reliable candidate samples. We then design a reward-guided active selection strategy to prioritize annotating the most representative samples within this set, maximizing the effectiveness of limited annotations. In addition, to balance pseudo-labels with scarce annotations, we introduce an adaptation objective that mitigates the imbalance and alleviates decision-boundary ambiguity. Extensive experiments demonstrate that AUTTA significantly improves performance and achieves state-of-the-art results under dual-shift scenarios.

## 1 Introduction

Universal Test-Time Adaptation (UTTA) (Schlachter & Yang, 2024; Schlachter et al., 2025; Li et al., 2023a) is a framework designed to effectively learn new classes while excluding old classes in the target domain during testing. It aims to tackle both *domain shift* and *class shift* without requiring access to source data, making it particularly suitable for real-world scenarios with evolving data distributions. However, when domain and class shifts occur simultaneously, pseudo-labeling methods tend to fail due to increased labeling errors. This paper introduces an active learning framework for UTTA, termed Active Universal Test-Time Adaptation (AUTTA), by incorporating a small number of human-annotated samples during testing. A practical example of AUTTA is auto-driving, where human intervention can be queried to provide small yet essential annotations of high-uncertainty scenarios, improving robustness. (See the AppendixA.1 for detail application settings.)

Nevertheless, AUTTA is not simply an enhancement of traditional active learning methods within the UTTA framework. This is because traditional active learning (Settles, 2009) methods typically select samples for annotation based on entropy or uncertainty. However, a previous study (You et al., 2019) shows that under dual transfer scenarios, these methods tend to select samples distributed with higher class shift, which not only reduces annotation effectiveness but also introduces significant biases in model training. To illustrate this, we conduct an in-depth analysis of the data distribution in the target domain and validate it through experiments, as shown in Fig. 1. The results show that samples in regions focused on class shift (Qiao et al., 2023; Zhao et al., 2025; Majee et al., 2024) or domain shift exhibit lower pseudo-label error rates, while those in dual-shift mixed regions have higher error

---

*Equal contribution
†Corresponding authors: Wei Feng (first), Fuyuan Hu

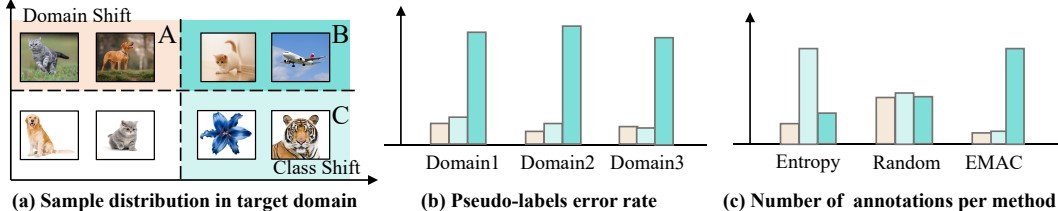

Figure 1: In the sample distribution under both class and domain dual shifts (a), mixed region B exhibits a higher degree of sample distribution confusion and a higher pseudo-label error rate (b). This results in more significant annotation effects. Traditional active learning methods fail to effectively select samples from region B for annotation (c), leading to wasted annotation costs. Best viewed in dark blue.

rates. Therefore, annotations should prioritize mixed regions to ensure their effectiveness. However, existing methods struggle to effectively cover these critical regions. Inspired by this, *We propose to expose samples in the mixed region under dual shifts and achieve effective annotation in AUTTA.*

In this paper, we introduce a novel method called Exposing Mixture and Annotating Confusion (EMAC) to select appropriate samples for annotation in AUTTA to improve the adaptation at test time. The sample annotation process in EMAC involves two stages. Exposing the mixture performs initial filtering to generate candidate samples with dual shifts. Annotating confusion refines the selection from the candidates. Specifically, we first design to expose mixed sample candidates using a Gaussian Mixture Model (GMM) to model the features after orthogonal decoupling, which prevents invalid labeling of other regions. Second, we introduce a reward-guided selection strategy, which can be theoretically connected to information gain in active learning (Gal et al., 2017; Sener & Savarese, 2018). Rather than relying purely on heuristic uncertainty measures, it quantifies the marginal reduction of entropy for old classes and the increased separability for new classes, aligning with the goal of minimizing adaptation error under dual shifts. With limited annotation budgets, this strategy prioritizes the most informative samples from the mixed regions, thereby maximizing annotation effectiveness. Finally, we design a clustering-based contrastive optimization objective to overcome the fuzzy decision boundary problem caused by limited annotated samples, combining annotation information with the pseudo-labels information balanced from unsupervised data. This effectively prevents the model from overfitting (Liu et al., 2023; Lin et al., 2020) and ensures consistent performance improvement. EMAC achieves state-of-the-art (SOTA) performance on both the DomainNet and VisDA-C datasets. Compared to existing UTTA methods, EMAC leverages human assistance to better learn inter-class distinctions in open-set scenarios. Additionally, unlike traditional active learning methods, EMAC excels in selecting the most representative and informative samples, effectively balancing annotation cost and model performance.

Our contributions are as follows:

(1) We propose a novel Universal Test Time Adaptation paradigm, Active Universal Test-Time Adaptation (AUTTA), which enhances the applicability and generalization of UTTA in open-class shift scenarios by effectively integrating additional information.

(2) We propose a new reward-guided active learning approach that ensures the informativeness of selected samples in open-set scenarios. Additionally, by incorporating a clustering-based contrastive objective, the approach improves robustness under the AUTTA protocol.

(3) We establish a benchmark for evaluating the AUTTA protocol, encompassing various types of domain shifts and class shift ratios, including common corruptions and style transfer. Our approach achieves state-of-the-art performance on the proposed benchmark.

## 2 RELATED WORK

### 2.1 UNIVERSAL TEST-TIME ADAPTATION

Universal Test-Time Adaptation (UTTA) (Schlachter et al., 2025) is designed to address the challenge of domain and class shifts that are prevalent in open-world environments. Unlike traditional TTA (Sun et al., 2020), which assumes the test data to be somewhat aligned with the source domain,

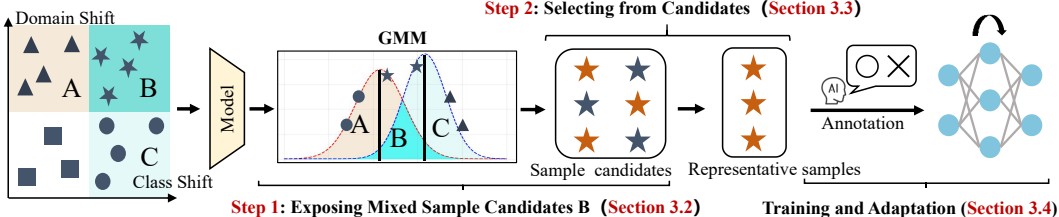

Figure 2: Sample selection and annotation process.

UTTA accounts for more complex situations where test data may include not only domain shifts (Lyu et al., 2023; Ni et al., 2025; Lyu et al., 2021) but also new, unknown categories or noise. This comprehensive approach ensures that the model can adapt in real time while handling diverse, uncertain and unpredictable test data. By focusing on such challenging scenarios, UTTA aims to improve the adaptability (Seabra et al., 2024; Qu et al., 2024a; Liu et al., 2025) and robustness of models in open-world settings, making it a more universal solution compared to conventional TTA. Specifically, TTAC (Su et al., 2022) employs distribution alignment at both global and category levels to facilitate test-time adaptation. OPTTT (Li et al., 2023b) combines self-training with prototype expansion to accommodate the strong OOD samples. Despite the progress made by methods such as TTAC and OPTTT, traditional approaches often struggle to address the need for sample diversity and the high uncertainty associated with real-world test-time scenarios, particularly without the incorporation of additional information.

### 2.2 ACTIVE LEARNING AND ATTA

Active Test-Time Adaptation (ATTA) (Gui et al., 2024) is a promising and lightweight domain adaptation method designed to optimize model generalization (Li et al., 2024) by integrating a small number of labeled samples during test time. Unlike most TTA methods, which can only handle small changes and heavily rely on heuristics and empirical studies (Pearl, 2009; Lin et al., 2022), ATTA combines active learning strategies and utilizes unlabeled data collected at test time to improve model performance through real-time sample selection and adaptation. Without access to source domain data, ATTA dynamically adjusts the model by detecting changes in the target domain, achieving better adaptation. However, the current ATTA algorithm inherits the limitations of Active Domain Adaptation (ADA) (Sener & Savarese., 2018; Rangwani et al., 2021; Xie et al., 2022) and Active Online Learning (AOL) (Cacciarelli & Kulahci, 2023). These methods typically focus on sample selection strategies in the context of drifting data streams (Zhou et al., 2021; Baier et al., 2021; Li et al., 2021) under domain shifts, without accounting for class shifts. As a result, in scenarios involving class shifts, the ATTA approach, similar to general active learning methods, often fails to address the need for both sample diversity and high uncertainty.

## 3 METHODOLOGY

### 3.1 PROBLEM DEFINITION

Active Universal Test-Time Adaptation (AUTTA) integrates active learning into the test-time adaptation process to address both class and domain shifts. In AUTTA, the model dynamically selects a small set of instances for annotation from an oracle (e.g., human annotators or self-supervised signals). The acquired labels are then incorporated into the adaptation process, providing additional supervision alongside pseudo-labeled data. Similar to UTTA, an AUTTA model is initialized from a pre-trained model $f(x;\theta)$ trained on the source domain $\mathcal{D}_S$, and is then deployed in an unlabeled target test domain $\mathcal{D}_{UT} = \{x_i\}_{i=1}^n$ for simultaneous testing and adaptation. At each step, the active learning algorithm queries the oracle for a limited number of labels, which are accumulated into a labeled target set $\mathcal{D}_{LT}$. Importantly, annotations may be partially acquired, reflecting the practical constraints of limited labeling budgets. The ultimate goal of AUTTA is to leverage these annotations to accurately classify old classes in the target domain while rejecting novel ones.

As illustrated in Fig. 2, our approach proceeds in two stages. First, we employ a Gaussian Mixture Model (GMM) (Qu et al., 2024b; Wang & Wang, 2024) to expose regions where samples are most likely to exhibit dual shifts, generating a candidate pool that captures the most informative samples. This coarse selection ensures that annotation resources are not wasted on low-value samples. Second, we evaluate the contribution of candidate samples to the model's ability to distinguish old from new classes, and select those with the greatest impact. This coarse-to-fine pipeline improves the efficiency of annotation under strict budget constraints. Finally, to address the class boundary ambiguity caused by scarce labeled data, we introduce a clustering-based contrastive loss that jointly leverages both annotations and pseudo-labels, thereby enhancing boundary clarity and improving robustness during test-time adaptation.

## 3.2 Exposing Mixed Sample Candidates

To tackle the mixture, we need to first expose the highly mixed regions between the new and old class samples and present candidates. Due to the unavailability of source data, we leverage the orthogonality and low-rank approximation properties of Singular Value Decomposition (SVD) to decompose the classifier $\mathbf{W}_{\text{cls}} \in \mathbb{R}^{C \times D}$ parameter space, which encapsulates source domain knowledge, into independent subspaces:

$$\mathbf{W}_{\text{cls}} = \mathbf{F}_{\text{known}} \Sigma \mathbf{F}_{\text{unknown}}^{\top}. \tag{1}$$

Eq. equation 1 enables the decoupling of source-known vector base $\mathbf{F}_{\text{known}}$ and the source-unknown vector base $\mathbf{F}_{\text{unknown}}$, effectively facilitating the separation of known and unknown information under class and domain shifts when source data is inaccessible. Let $\mathbf{z} \in \mathbb{R}^{B \times D}$ represent the normalized features of a batch of target data $\mathcal{D}_{\text{UT}}^{b}$:

$$\mathbf{z} = f(\mathcal{D}_{\text{UT}}^{b}; \theta). \tag{2}$$

Then, $\mathbf{z}$ is projected onto $\mathbf{F}_{\text{known}}$ and $\mathbf{F}_{\text{unknown}}$ as follows:

$$
\begin{aligned}
\mathbf{z}_{\text{known}} &= \mathbf{z}^{B \times (1:C)} \cdot \mathbf{F}_{\text{known}}, \\
\mathbf{z}_{\text{unknown}} &= \mathbf{z}^{B \times (C+1:D)} \cdot \mathbf{F}_{\text{unknown}}^{C+1:D},
\end{aligned}
\tag{3}
$$

where, $\|\mathbf{z}_{\text{known}}\|_2^2 + \|\mathbf{z}_{\text{unknown}}\|_2^2 = 1$.

Intuitively, through feature decomposition, we can estimate the distribution of unknown information in $\|\mathbf{z}_{\text{unknown}}\|_2^2$. Our observations show that the empirical distribution of $\|\mathbf{z}_{\text{unknown}}\|_2^2$ exhibits a bimodal pattern, with two distinct peaks, typically indicating the presence of old classes and new classes of data. Therefore, we use a bimodal Gaussian mixture model to estimate the distribution of $\|\mathbf{z}_{\text{unknown}}\|_2^2$, where the component with a lower mean represents old classes data, and the component with a higher mean represents new classes data. Specifically, we use $\mu_{\text{old}}$ and $\mu_{\text{new}}$ to represent the expectations of old classes and new classes data, with $\mu_{\text{old}} < \mu_{\text{new}}$. The Probability Density Function (PDF) is as follows:

$$p(\mathbf{z}_{\text{unknown}}) = \pi \mathcal{N}(\mu_{\text{old}}, \sigma_{\text{old}}^2) + (1 - \pi)\mathcal{N}(\mu_{\text{new}}, \sigma_{\text{new}}^2). \tag{4}$$

where, $\pi = 0.5$. We define three types of sample feature regions using $\mu_{\text{old}}$ and $\mu_{\text{new}}$ as follows:

$$\mathcal{X}_{\text{old}} = \mathcal{D}_{\text{UT}}[i| \sum\nolimits_{i=1}^{n} \mathbf{1}(\mathbf{z}_{\text{unknown}}^i < \mu_{\text{old}})], \tag{5}$$

$$\mathcal{X}_{\text{new}} = \mathcal{D}_{\text{UT}}[i| \sum\nolimits_{i=1}^{n} \mathbf{1}(\mathbf{z}_{\text{unknown}}^i > \mu_{\text{new}})], \tag{6}$$

$$\mathcal{X}_{\text{mix}} = \mathcal{D}_{\text{UT}}[i| \sum\nolimits_{i=1}^{n} \mathbf{1}(\mu_{\text{old}} < \mathbf{z}_{\text{unknown}}^i < \mu_{\text{new}})]. \tag{7}$$

Using the GMM, three distinct regions are identified. Samples from the two peripheral regions ($\mathcal{X}_{\text{old}}$ and $\mathcal{X}_{\text{new}}$) are assumed to experience higher levels of domain or class shifts. However, selecting samples with them requires more annotations and fails to address the dual shift confusion issue effectively. Alternatively, pseudo-labeling can partially mitigate this issue but lacks robustness. In contrast, samples from the mixed region $\mathcal{X}_{\text{mix}}$ exhibit significant confusion due to the dual shift effect. Selecting samples from this region for annotation offers higher potential benefits, as these samples contribute more to resolving the mixture caused by overlapping shifts. Therefore, we select samples from the mixed region as candidate samples to maximize annotation effectiveness, thereby enhancing the model's adaptability and performance under dual-shift conditions. Further discussion of GMM details in the appendixA.2.

## 3.3 SELECTING FROM CANDIDATES

Although the above method provides a set of sample candidates, distributional bias within these candidates can result in severe annotation imbalances if samples are randomly selected. To address this issue, we propose a reward-guided active selection strategy, which prioritizes annotating representative samples that contribute most to model optimization.

Formally, we design a Max-Min Entropy (MME) objective function:

$$\mathcal{L}_{\mathrm{MME}} = \Gamma_{\mathrm{old}} + \Gamma_{\mathrm{new}}, \tag{8}$$

where $\Gamma_{\mathrm{old}}$ and $\Gamma_{\mathrm{new}}$ quantify the marginal information gain of annotated samples for old and new classes, respectively. This design is theoretically connected to active learning bounds: maximizing entropy reduction of old classes improves the retention of known decision boundaries, while maximizing entropy increase for new classes enhances the margin between known and novel categories. Thus, $\mathcal{L}_{\mathrm{MME}}$ can be interpreted as minimizing an empirical generalization error bound under limited annotation budgets.

Specifically, $\Gamma_{\mathrm{old}}$ measures how annotations improve the retention of old-class knowledge, i.e., the model's increased confidence in recognizing old classes:

$$\Gamma_{\mathrm{old}} = \sum_{i=1} H(f(\mathcal{X}_{\mathrm{old}}, \theta^t)) - H(f(\mathcal{X}_{\mathrm{old}}, \theta^{t-1})), \tag{9}$$

where $H(\cdot)$ denotes the entropy function. Conversely, $\Gamma_{\mathrm{new}}$ measures the extent to which annotations help exclude new classes by increasing the model's confidence in rejecting them:

$$\Gamma_{\mathrm{new}} = \sum_{i=1} H(f(\mathcal{X}_{\mathrm{new}}, \theta^{t-1})) - H(f(\mathcal{X}_{\mathrm{new}}, \theta^t)). \tag{10}$$

To further balance annotation efforts between old and new classes, we define average reward values for both types of annotated samples:

$$R'_{\mathrm{old}} = \frac{1}{|\mathcal{D}_{\mathrm{LT}}^{\mathrm{old}}|} \cdot (\omega_{\mathrm{old}} \cdot \Gamma_{\mathrm{old}} + (1 - \omega_{\mathrm{new}}) \cdot \Gamma_{\mathrm{new}}), \tag{11}$$

$$R'_{\mathrm{new}} = \frac{1}{|\mathcal{D}_{\mathrm{LT}}^{\mathrm{new}}|} \cdot ((1 - \omega_{\mathrm{old}}) \cdot \Gamma_{\mathrm{old}} + \omega_{\mathrm{new}} \cdot \Gamma_{\mathrm{new}}), \tag{12}$$

where $\mathcal{D}_{\mathrm{LT}}^{\mathrm{old}} = \{(x_i^{\mathrm{old}}, y_i^{\mathrm{old}})\}_{i=1}^n$ and $\mathcal{D}_{\mathrm{LT}}^{\mathrm{new}} = \{(x_i^{\mathrm{new}}, y_i^{\mathrm{new}})\}_{i=1}^n$ denote labeled samples from old and new classes, respectively. Here, $\omega_{\mathrm{old}}$ and $\omega_{\mathrm{new}}$ are weighting factors reflecting the contribution ratio of old and new classes. These weights can be interpreted as balancing terms ensuring that the variance of the empirical risk estimator does not dominate in either direction.

To stabilize selection, we further apply an exponential moving average (EMA) update:

$$R_{\mathrm{old}} = \alpha \cdot R'_{\mathrm{old,t}} + (1 - \alpha) R'_{\mathrm{old,t-1}}, \tag{13}$$

$$R_{\mathrm{new}} = \alpha \cdot R'_{\mathrm{new,t}} + (1 - \alpha) R'_{\mathrm{new,t-1}}, \tag{14}$$

where $\alpha$ is the smoothing factor. This temporal smoothing aligns with theoretical analyses of regret minimization in online learning, ensuring that short-term fluctuations do not bias the annotation strategy.

Finally, a selection rule is applied to dynamically guide annotation. We select the highest-entropy old-class samples and the lowest-entropy new-class samples from the mixed region:

$$\begin{cases} \mathcal{D}_{\mathrm{LT}}^{\mathrm{old}} = \{ \underset{x_{\mathrm{old}} \in \mathcal{X}_{\mathrm{mix}}}{\arg\max} \ H(x_{\mathrm{old}})\} \cup \mathcal{D}_{\mathrm{LT}}^{\mathrm{old}}, & R_{\mathrm{old}}^t > R_{\mathrm{new}}^t, \\ \mathcal{D}_{\mathrm{LT}}^{\mathrm{new}} = \{ \underset{x_{\mathrm{new}} \in \mathcal{X}_{\mathrm{mix}}}{\arg\min} \ H(x_{\mathrm{new}})\} \cup \mathcal{D}_{\mathrm{LT}}^{\mathrm{new}}, & \text{otherwise}, \end{cases} \tag{15}$$

where $|\mathcal{D}_{\mathrm{LT}}| = |\mathcal{D}_{\mathrm{LT}}^{\mathrm{old}}| + |\mathcal{D}_{\mathrm{LT}}^{\mathrm{new}}|$ denotes the total annotation budget.

In summary, the proposed reward-guided strategy can be interpreted as an entropy-based information gain maximization mechanism, theoretically reducing adaptation error under limited annotation budgets.

### 3.4 BALANCING TRUE-LABELS AND PSEUDO-LABELS OPTIMIZATION

When limited annotated samples are available, relying solely on annotated information for unsupervised domain adaptation is impractical. Therefore, pseudo-labels and other information must be used simultaneously. However, improper optimization may cause imbalanced learning, leading to class boundary ambiguity and model collapse. To address this issue, we propose a clustering-based contrastive loss that enables the model to adaptively learn by balancing the use of pseudo-labels and annotated information.

First, we perform clustering using high-confidence pseudo-label features $\mathcal{F}_{\mathrm{p}} = f(\mathcal{D}_{\mathrm{UT}}) = \{(\mathbf{u}_i, \hat{y}_i = c)\}$ and annotated features $\mathcal{F}_{\mathrm{a}} = f(\mathcal{D}_{\mathrm{LT}}) = \{(\mathbf{v}_j, y_j = c)\}$ to obtain class prototype $\mathbf{p}_c$:

$$\mathbf{p}_c = \frac{1}{\mid \mathcal{F}_{\mathrm{p}} \mid \mid \mathcal{F}_{\mathrm{a}} \mid} \left( \sum_{\mathbf{u}_i \in \mathcal{F}_{\mathrm{p}}} \mathbf{u}_i + \sum_{\mathbf{v}_i \in \mathcal{F}_{\mathrm{a}}} \mathbf{v}_i \right). \tag{16}$$

The cluster enhances the model's robustness and prevents overfitting and class boundary ambiguity that may arise when relying solely on annotation. The contrastive loss should enhance learning by minimizing intra-class distances, clustering pseudo-labeld samples within the same class and reinforcing their attraction to class prototypes. Simultaneously, it maximizes inter-class distances by separating known and unknown class samples and ensuring unknown samples remain dispersed and distant from class prototypes to prevent misclassification.

To achieve this, we first extract features of old and new classes, denoted as $\mathcal{F}_{\mathrm{old}}$ and $\mathcal{F}_{\mathrm{new}}$. Next, data augmentation is applied to each sample in both sets, transforming them into feature space and producing $\mathcal{R}_{\mathrm{old}}$ and $\mathcal{R}_{\mathrm{new}}$. For each augmented sample pair, we associate the corresponding class prototype set $\mathcal{P} = \{\mathbf{p}_c\}_{c=1}^C$. Map $\{\mathcal{F}_{\mathrm{old}}, \mathcal{R}_{\mathrm{old}}, \mathcal{P}\}$ and $\{\mathcal{F}_{\mathrm{new}}, \mathcal{R}_{\mathrm{new}}, \mathcal{P}\}$ to a low-dimensional projection space to form a unified set of feature representations $\mathcal{Q}$. By calculating the clustering-contrastive loss based on these representations, annotated information can effectively guide high-confidence pseudo-labels, preventing class boundary ambiguity caused by model overfitting:

$$\mathcal{L}_{\mathrm{c}} = \sum_{i \in \mathcal{I}_{\mathrm{old}}} \frac{-1}{|\mathcal{Q}(i)|} \sum_{p \in \mathcal{Q}(i)} \log \frac{\exp(s_{ip})}{S(i)}, \tag{17}$$

where $\mathcal{I}_{\mathrm{old}}$ and $\mathcal{I}_{\mathrm{new}}$ represent the set of indices of the old and new classes,

$$S(i) = \sum_{w \in \mathcal{N}(i)} \exp(s_{iw}) + \sum_{j \in \mathcal{I}_{\mathrm{old}}} \exp(s_{ij}). \tag{18}$$

$\mathcal{N}(i) = \mathcal{I}_{\mathrm{new}} \cup \mathcal{I}_{\mathrm{old}}$ represents the negative sample set. Moreover, $s_{ij}$ is the cosine similarity. Finally, the overall objective $\mathcal{L}$ for EMAC at the $t$-th iteration is:

$$\mathcal{L} = \mathcal{L}_{\mathrm{MME}} + \mathcal{L}_{\mathrm{c}}. \tag{19}$$

EMAC reduces classification errors by selecting appropriate samples for annotation and optimizing the balance between true labels and pseudo-labels for adaptation.

In summary, we first expose mixed sample candidates by modeling the orthogonally decomposed features with GMM to identify regions with the highest annotation validity, thereby avoiding the waste of limited annotation resources. Next, we adopt a reward-guided active selection strategy that quantifies the marginal information gain of each annotated sample with respect to the model's ability to separate old and new classes. By explicitly linking the reward to entropy reduction and class separability, this strategy ensures that, under a limited annotation budget, the most informative and representative samples are prioritized. Finally, we introduce Balancing True-labels and Pseudo-labels Optimization by designing a clustering-based contrastive loss, which enhances intra-class cohesion and inter-class separation. This significantly improves the model's discriminative ability under label scarcity and alleviates the decision boundary ambiguity caused by limited labeled data.

In Algorithm 1, we first expose mixed sample candidates by modeling the orthogonally decomposed features with GMM to identify regions with the highest annotation validity, thereby avoiding the waste of limited annotation resources. Next, we adopt a reward-guided active selection strategy that quantifies the marginal information gain of each annotated sample with respect to the model's ability to separate old and new classes. By explicitly linking the reward to entropy reduction and class separability, this strategy ensures that, under a limited annotation budget, the most informative and

---

**Algorithm 1** EMAC.

---

**Input:** Pre-trained model $f(x; \theta)$, classifier $\mathbf{W}_{\text{cls}}$, unlabele d target data $\mathcal{D}_{\text{UT}}$.

1: **for** Each batch of target data $\mathcal{D}_{\text{UT}}^b$ **do**
2:     Obtain a batch of features $\mathbf{z}$ on $f(\mathcal{D}_{\text{UT}}^b; \theta)$
     *#Exposing Mixed Sample Candidates*
3:     Obtain source-unknown features $\mathbf{z}_{\text{unknown}}$ using SVD (Eq. equation 3)
4:     Obtain candidates $\mathcal{X}_{\text{mix}}$ and $\mathcal{X}_{\text{old}}$, $\mathcal{X}_{\text{new}}$ on $\mathbf{z}_{\text{unknown}}$ and $\mathcal{D}_{\text{UT}}^b$ using GMM (Eq. equation 7)
     *#Selecting from Candidates*
5:     Compute $\mathcal{L}_{\text{MME}}$ on $\mathcal{X}_{\text{old}}, \mathcal{X}_{\text{new}}$ (Eq. equation 8)
6:     Compute Reward $R_{\text{old}}, R_{\text{new}}$ on $\mathcal{L}_{\text{MME}}$ (Eq. equation 14)
7:     Update annotation set $\mathcal{D}_{\text{LT}}$ on candidates $\mathcal{X}_{\text{mix}}$ and conditions $R_{\text{old}}, R_{\text{old}}$ (Eq. equation 15)
     *#Balancing True-labels and Pseudo-labels*
8:     Cluster class prototype $\mathcal{L}_{\text{c}}$ on $\mathcal{D}_{\text{LT}}$ and $\mathcal{D}_{\text{UT}}^b$ (Eq. equation 16)
9:     Compute $\mathcal{L}_c$ on $\mathcal{X}_{\text{old}}, \mathcal{X}_{\text{new}}$ and $\mathbf{p}_c$ (Eq. equation 17)
10:    Training and adaptation using $\mathcal{L}_{\text{MME}}$ and $\mathcal{L}_c$
11: **end for**

---

representative samples are prioritized. Finally, we introduce Balancing True-labels and Pseudo-labels Optimization by designing a clustering-based contrastive loss, which enhances intra-class cohesion and inter-class separation. This significantly improves the model's discriminative ability under label scarcity and alleviates the decision boundary ambiguity caused by limited labeled data.

## 4 EXPERIMENT

**Datasets.** We evaluate EMAC on two DA datasets: DomainNet and VisDA-C. **DomainNet** comprises approximately 0.6 million images spanning 345 classes across three domains: painting (P), real (R), and sketch (S). All 345 classes are present in each domain. We evaluate COMET across all six domain shifts: P→R, P→S, R→P, R→S, S→P, and S→R. **VisDA-C** includes 12 object classes and poses a challenging domain shift, transitioning from synthetic 2D renderings of 3D models to real-world images from the Microsoft COCO dataset. Consistent with prior work, we use the first domain as the source domain for evaluation.

**Evaluation Metric.** We adopt the evaluation metrics from OPTTT (Li et al., 2023a), which are designed for open-set classification tasks. UTTA focuses on two key aspects: the accuracy of source domain classes and the effectiveness of rejecting private target domain classes. Specifically, $A_O$ measures the accuracy of old target classes, while $A_N$ evaluates the rejection accuracy of new target classes. Their harmonic mean, $A_H$, offers a balanced assessment of overall predictive performance.

### 4.1 COMPARISON TEST-TIME ADAPTATION METHODS

We benchmark the following test-time training methods. TEST directly evaluates the source domain model on the test data without adaptation. BN (Ioffe & Szegedy, 2015) updates batch normalization statistics using the test data for test-time adaptation. TTT++ (Sun et al., 2020) minimizes the Frobenius norm between mean covariances to align source and target domain distributions. TENT (Wang et al., 2021) adapts the model by updating batch normalization affine parameters through entropy minimization on test data. SHOT (Liang et al., 2020) employs entropy minimization and self-training, assuming a class-balanced target domain and encouraging uniform prediction distributions. BiTTA (Lee et al., 2025) introduces binary correctness feedback for test-time adaptation, correcting unstable predictions through lightweight human-in-the-loop signals. TTAC (Su et al., 2022) performs distribution alignment at both global and category levels to enhance test-time training. SimATTA (Gui et al., 2024) leverages active learning to annotate a small number of target samples, improving model generalization. EATTA (Wang & Ding, 2025) proposes an effortless active labeling mechanism to aid long-term test-time adaptation by selecting samples based on confidence-driven criteria. OPTTT (Li et al., 2023a) addresses open-world test-time training by explicitly modeling class shifts for the first time. Finally, we introduce EMAC, which integrates active learning with adaptation to open-world scenarios involving dual shifts. For fair comparison, we maintain consistent pipelines across all baseline methods.

Table 1: Active Universal Test-Time Adaptation results on DomainNet. All numbers are in %. (The best options in methods are shown in bold and ∗ indicates the pseudo-update version. The unit of GPU times is in seconds.)

| Task | Method | S2R | | | S2P | | | P2R | | | P2S | | | R2S | | | R2P | | | Avg. | |
|---|---|---|---|---|---|---|---|---|---|---|---|---|---|---|---|---|---|---|---|---|---|
| | | $A_O$ | $A_N$ | $A_H$ | $A_O$ | $A_N$ | $A_H$ | $A_O$ | $A_N$ | $A_H$ | $A_O$ | $A_N$ | $A_H$ | $A_O$ | $A_N$ | $A_H$ | $A_O$ | $A_N$ | $A_H$ | $A_H$ | Sec. |
| TTA | TEST | 34.5 | **78.1** | 47.8 | 20.0 | **76.2** | 32.0 | 45.7 | **77.8** | 57.6 | 25.2 | **84.4** | 38.8 | 25.2 | **82.5** | 38.6 | 33.7 | **81.1** | 47.6 | 43.7 | 392 |
| | BN | 36.5 | 70.2 | 48.0 | 24.6 | 61.9 | 35.2 | 48.0 | 68.3 | 56.4 | 30.7 | 72.1 | 43.1 | 28.6 | 70.5 | 40.7 | 36.4 | 69.3 | 47.7 | 45.2 | 577 |
| | TENT | 36.8 | 68.4 | 47.9 | 25.8 | 61.4 | 36.3 | 49.6 | 66.8 | 56.9 | 33.8 | 68.1 | 45.2 | 31.0 | 67.4 | 42.5 | 36.1 | 70.2 | 47.6 | 46.1 | 479 |
| | SHOT | 37.5 | 66.9 | 48.1 | 33.9 | 51.1 | 40.8 | 47.5 | 68.5 | 56.1 | 28.6 | 80.4 | 42.2 | 27.8 | 76.3 | 40.8 | 34.4 | 80.3 | 48.2 | 46.0 | 564 |
| UTTA | TTAC | 40.1 | 62.8 | 48.9 | 40.6 | 56.3 | 47.2 | 36.8 | 82.6 | 51.0 | 31.8 | 65.2 | 42.7 | 29.7 | 66.2 | 41.0 | 37.5 | 74.6 | 49.9 | 46.7 | 651 |
| | OPTTT | 45.7 | 61.4 | 52.4 | 40.2 | 59.6 | 48.0 | 52.2 | 63.7 | 57.4 | 33.1 | 68.6 | 44.6 | 34.1 | 68.0 | 46.1 | 38.9 | 78.3 | 51.9 | 49.8 | 697 |
| ATTA | SimATTA | 41.3 | 58.6 | 48.5 | 36.9 | 57.1 | 44.8 | 49.3 | 60.1 | 54.2 | 28.9 | 78.1 | 42.2 | 31.5 | 70.5 | 43.5 | 35.5 | 72.6 | 48.7 | 47.1 | 747 |
| | EATTA | 41.5 | 58.9 | 48.7 | 37.8 | 57.9 | 45.7 | 49.9 | 61.0 | 54.9 | 30.1 | 78.6 | 43.5 | 32.4 | 71.3 | 44.6 | 36.4 | 73.4 | 48.7 | 47.7 | 738 |
| | BiTTA | 37.5 | 68.4 | 48.4 | 27.9 | 61.7 | 38.4 | 49.1 | 67.9 | 57.0 | 31.7 | 75.2 | 44.6 | 32.3 | 72.8 | 44.7 | 37.9 | 73.6 | 50.0 | 47.2 | 687 |
| AUTTA | EMAC | 47.7 | 66.4 | 55.5 | 42.3 | 57.3 | 48.7 | 53.5 | 66.8 | 59.4 | 34.9 | 71.6 | 46.9 | **35.9** | 72.0 | **47.9** | 41.7 | 79.6 | 54.7 | 52.2 | 735 |
| | EMAC* | **48.2** | 67.2 | **56.1** | **43.9** | 57.7 | **49.9** | **53.8** | 67.4 | **59.8** | **36.6** | 72.8 | **48.7** | 35.2 | 72.6 | 47.4 | **42.3** | 80.6 | **55.5** | **53.1** | 779 |

Table 2: AUTTA results on VisDA-C.

| Method | NOISE | | | MNIST | | | SVHN | | |
|---|---|---|---|---|---|---|---|---|---|
| | $A_O$ | $A_N$ | $A_H$ | $A_O$ | $A_N$ | $A_H$ | $A_O$ | $A_N$ | $A_H$ |
| TEST | 48.2 | **100.0** | 65.1 | 48.6 | **98.8** | 65.1 | 49.4 | **98.5** | 64.9 |
| BN | 52.6 | **100.0** | 68.9 | 50.1 | 82.3 | 62.3 | 58.5 | 86.2 | 69.7 |
| TENT | 58.8 | 98.9 | 73.8 | 46.3 | 68.2 | 55.1 | 51.0 | 81.5 | 62.7 |
| SHOT | 62.1 | **100.0** | 76.6 | 48.2 | 36.0 | 41.2 | 56.5 | 62.8 | 59.5 |
| TTAC | 59.2 | **100.0** | 74.4 | 52.0 | 67.4 | 58.7 | 57.9 | 66.3 | 61.4 |
| OPTTT | 63.9 | **100.0** | 77.8 | 64.2 | 90.7 | 75.2 | 64.5 | 74.8 | 69.2 |
| SimATTA | 61.1 | **100.0** | 75.8 | 56.7 | 69.5 | 62.5 | 59.5 | 67.8 | 63.4 |
| EATTA | 62.0 | **100.0** | 76.5 | 60.2 | 78.1 | 68.0 | 63.1 | 72.0 | 67.3 |
| BiTTA | 62.8 | **100.0** | 77.1 | 51.5 | 68.0 | 58.6 | 60.3 | 74.0 | 66.5 |
| EMAC | 66.4 | **100.0** | 79.8 | 66.5 | 92.8 | 77.4 | 69.4 | 75.2 | 72.3 |
| EMAC* | **67.2** | **100.0** | **80.4** | **66.8** | 94.7 | **78.3** | **70.6** | 76.1 | **73.2** |

Table 3: Comparisons to AL baselines in AUTTA.

| DomainNet | S2R | S2P | P2R | P2S | R2S | R2P | Avg. | VisDA-C |
|---|---|---|---|---|---|---|---|---|
| Random ($\mathcal{B} = 1000$) | 48.2 | 44.6 | 53.6 | 43.9 | 42.3 | 48.7 | 46.8 | 66.7 |
| Entropy ($\mathcal{B} = 1000$) | 47.6 | 43.9 | 52.8 | 44.6 | 41.5 | 47.4 | 46.3 | 65.4 |
| Kmeans ($\mathcal{B} = 1000$) | 48.2 | 44.3 | 52.1 | 45.9 | 41.3 | 48.6 | 46.7 | 66.8 |
| CLUE ($\mathcal{B} = 1000$) | 44.6 | 47.5 | 55.7 | 40.3 | 39.8 | 50.7 | 46.4 | 67.3 |
| Coreset ($\mathcal{B} = 1000$) | 49.8 | 46.8 | 58.3 | 44.4 | 47.1 | 48.6 | 49.4 | 68.7 |
| BADGE ($\mathcal{B} = 1000$) | 50.4 | 47.3 | 54.6 | 45.8 | 46.2 | 52.4 | 49.5 | 68.2 |
| SimATTA ($\mathcal{B} = 1000$) | 48.5 | 44.8 | 54.2 | 42.2 | 43.5 | 48.7 | 47.1 | 67.9 |
| EMAC ($\mathcal{B} = 800$) | 54.5 | 47.1 | 58.1 | 45.4 | 46.7 | 53.6 | 50.8 | 73.1 |
| EMAC ($\mathcal{B} = 1000$) | 55.5 | 48.7 | 59.4 | 46.9 | 47.9 | 54.7 | 52.2 | 76.5 |
| EMAC ($\mathcal{B} = 1500$) | **56.3** | **48.9** | **60.8** | **47.8** | **48.7** | **55.6** | **53.1** | **78.2** |

## 4.2 COMPARISONS BETWEEN EMAC AND THE SOTAS

**Noise Corrupted Target Domain.** We first evaluate EMAC under noise-corrupted target domains, using painting (P), real (R), and sketch (S) from DomainNet as the source domain and adapting to (R, S, P), respectively. Results in Tab. 18 reveal several trends. Direct testing does not always produce the worst performance, as the model is more biased by class shift than by pure domain shift and therefore often predicts unseen classes as unknown rather than collapsing uniformly. TTA baselines (BN, TENT, SHOT, TTAC) improve robustness through distribution alignment but still struggle to disentangle domain shift from new-class uncertainty. OPTTT better handles open-world conditions via prototype extension, while BiTTA provides binary correction signals but remains too coarse to identify informative samples under universal shift. Active-TTA methods such as SimATTA and EATTA benefit from limited labeling but rely heavily on confidence-based scoring, which is unreliable when class and domain shifts co-occur, leading to suboptimal querying. EMAC explicitly exposes mixed samples and performs reward-guided querying, achieving the best performance across all settings. Its reinforced variant EMAC∗ further prevents negative updates from sparse annotations. Additional results and GPU-time analysis are provided in Appendix A.3.3.

**Style Transfer Target Domain.** We further evaluate VisDA-C to demonstrate the effectiveness of the style transfer target domain. The results are presented in Tab.2. We draw similar conclusions from the results as with common corruption. Additionally, the number of categories in VisDA-C is much smaller than in DomainNet. The results show that all target domain adaptations in VisDA-C outperform those in DomainNet. It can be inferred that when the number of annotated samples far exceeds the number of categories, the imbalance in optimizing limited annotation costs and pseudo-labels will be reduced. Therefore, our proposed method consistently outperforms competing methods in both complex and simple open environments.

## 4.3 ADDITIONAL ANALYSIS

**Comparison of Active Learning Methods.** To validate the superiority of active learning in AUTTA and to ensure a fair comparison, we wrap all active learning methods (Random, Entropy (Wang & Shang, 2014), Kmeans, CLUE (Prabhu et al., 2021), Coreset (Sener & Savarese., 2018), BADGE (Ash et al., 2019), simATTA (Gui et al., 2024)) in the TENT framework and compare them with EMAC, ensuring they operate under the same conditions and share the same model backbone.

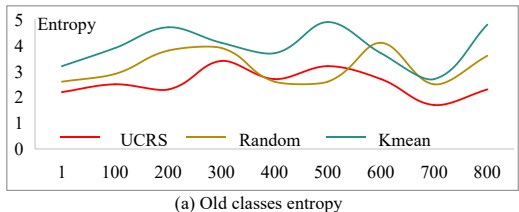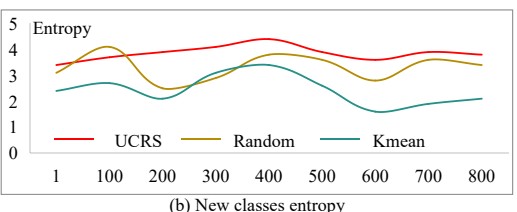

(a) Old classes entropy  (b) New classes entropy

Figure 3: The degree of annotated samples on the model's ability to distinguish between new and old classes.

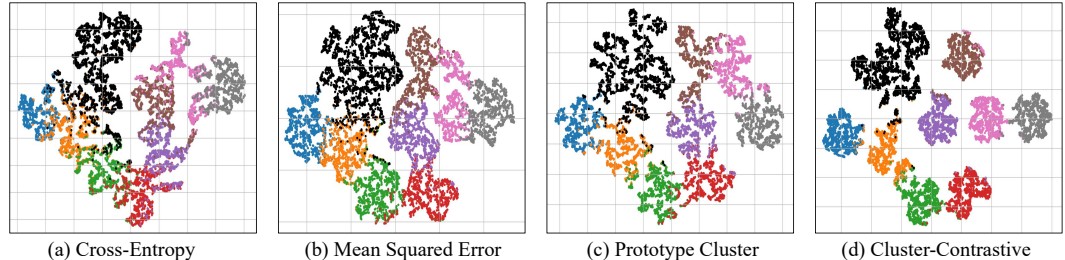

(a) Cross-Entropy  (b) Mean Squared Error  (c) Prototype Cluster  (d) Cluster-Contrastive

Figure 4: T-SNE visualization on DomainNet. Black dots are new class examples, while other colors are old class examples. The colors indicate different classes.

As shown in Table 3, the results demonstrate that incorporating a limited number of label test instances $\mathcal{B}$ significantly improves performance across test domains and class shifts. However, EMAC outperforms or is competitive with other active learning methods due to the higher effectiveness of the labeled samples.

**Error Rate in Mix-Gaussian and Entropy.** As shown in Table.4, to validate the rationale for introducing GMM and selecting $\mathcal{X}_{\mathrm{mix}}$, we first use the sample entropy (Wang & Shang, 2014) method to divide samples into three regions (EN. $< 0.25$, $0.25 < $ EN. $< 0.75$, EN. $> 0.75$) based on fixed thresholds (0.25 and 0.75), simulating GMM's segmentation. Next, We randomly select 1,000 samples from the three regions divided using the EMSC and entropy methods in target domains (Painting, Real, Sketch), and compare the predictions with the ground truth. The results indicate that with the EMSC method, the error rate in $\mathcal{X}_{\mathrm{mix}}$ is significantly higher, while the error rates in $\mathcal{X}_{\mathrm{old}}$ and $\mathcal{X}_{\mathrm{new}}$ remain negligible. In contrast, the entropy method shows non-negligible error rates across all regions.

Table 4: Error rate (%) in gmm and entropy.

| Domain | $\mathcal{X}_{\mathrm{old}}$ | EN.$< 0.25$ | $\mathcal{X}_{\mathrm{new}}$ | EN.$> 0.75$ | $\mathcal{X}_{\mathrm{mix}}$ | $0.25<$EN.$<0.75$ |
|---|---|---|---|---|---|---|
| Painting | 12.4 | 15.4 | 14.3 | 32.5 | 82.6 | 77.8 |
| Real | 10.2 | 15.6 | 11.5 | 34.6 | 86.4 | 71.2 |
| Sketch | 11.4 | 12.8 | 13.8 | 37.1 | 84.0 | 68.8 |

**Visualization Analysis.** To demonstrate the representation adaptation effectiveness of different loss functions, we utilize t-SNE (der Maaten & Hinton, 2008) to reduce feature dimensionality for visualization. In Fig. 4, we compare commonly used loss functions, including cross-entropy loss, mean squared error loss, prototype learning loss, and Cluster-Contrastive (OURS). Cluster-Contrastive achieves a more distinct separation between old classes (colored) and new classes (black) samples. Additionally, we observe that it enhances the separation within old class samples, highlighting its effectiveness in improving feature representation. These results demonstrate that cluster-contrastive loss $\mathcal{L}_c$ effectively balances the use of scarce annotated information and pseudo-labels, enabling the model to overcome class boundary ambiguity and achieve better training and adaptation.

**Sensitivity to Small Batch Sizes.** We investigate the sensitivity of EMAC to small batch sizes ($\leq 8$). For such small batches, naive per-batch computation often becomes unstable during GMM fitting, leading to noisy uncertainty estimation. To address this, we adopt a sliding window buffer and Exponential Moving Average(EMA) smoothing. The sliding window buffer accumulates data over multiple iterations, providing more reliable GMM fitting, while EMA smooths uncertainty estimates by considering both current and past values, reducing short-term fluctuations. The results in Table 5 show a significant performance gain on the S2R (DomainNet) dataset after applying these techniques,

Table 5: Sensitivity to Small Batch Sizes.

| Batch Size | Method | $A_O$ | $A_N$ | $A_H$ |
|---|---|---|---|---|
| 1 | EMAC (naive) | 12.5 | 24.3 | 16.3 |
| 1 | EMAC (w/ window) | 39.0 | 60.1 | 47.3 |
| 4 | EMAC (naive) | 19.5 | 32.3 | 20.1 |
| 4 | EMAC (w/ window) | 40.1 | 61.9 | 48.6 |
| 8 | EMAC (naive) | 27.8 | 39.5 | 32.5 |
| 8 | EMAC (w/ window) | 41.3 | 62.7 | 49.1 |

Table 6: Ablation study.

| EMSC | SC | BTPO | DomainNet | | | VisDA-C | | |
|---|---|---|---|---|---|---|---|---|
| | | | $A_O$ | $A_N$ | $\bar{A}_H$ | $A_O$ | $A_N$ | $\bar{A}_H$ |
| ✓ | - | - | 30.1 | **80.2** | 43.7 | 54.3 | 81.8 | 65.0 |
| - | ✓ | - | 41.0 | 54.9 | 47.1 | 60.1 | 82.3 | 69.4 |
| - | - | ✓ | 42.5 | 54.7 | 47.9 | 61.4 | 84.6 | 71.2 |
| ✓ | ✓ | - | 43.6 | 56.3 | 49.1 | 61.8 | 85.1 | 71.7 |
| - | ✓ | ✓ | 44.8 | 58.9 | 50.8 | 64.0 | 85.3 | 73.1 |
| ✓ | ✓ | ✓ | **45.5** | 61.2 | **52.2** | **66.9** | **89.3** | **76.5** |

demonstrating their effectiveness in stabilizing performance under small batch sizes. Full details are provided in the AppendixA.3.7.

**Ablation Study.** We analyze the effectiveness of the three components: Exposing Mixed Sample Candidates (EMSC), Using Selecting from Candidates (SC), and Balancing True-labels and Pseudo-labels Optimization (BTPO), as detailed in Tab.6. When used individually, each component shows limited effectiveness. For example, SC depends on EMSC to identify annotating regions, ensuring labeled sample validity, while BTPO relies on annotated labels to update class prototypes and prevent boundary blurring. SC leverages a max-min entropy method to refine boundaries between new and old classes by selecting high-contribution samples. Meanwhile, BTPO strengthens boundaries among old classes and further separates new and old classes by updating prototypes. Additionally, BTPO enhances EMSC's discrimination ability, improving the quality of samples for SC and ensuring stable, meaningful prototype updates. This synergy forms a positive feedback loop, progressively boosting model performance and adaptability, Further Additional Experiments Analysis in the appendixA.3.

# 5 CONCLUSION

This paper presents Active Universal Test-Time Adaptation (AUTTA), a novel framework that effectively handles both domain and class shifts with minimal human annotation. By integrating active learning and universal test-time adaptation, AUTTA improves model adaptability in dynamic environments. The core innovation, EMAC, uses Gaussian Mixture Models (GMM) for sample selection and a reward-guided strategy to optimize annotation efficiency. Experiments on DomainNet and VisDA-C show that AUTTA outperforms existing methods, demonstrating its effectiveness in dual-shift scenarios. While promising, AUTTA still relies on human annotations, which can be a limitation in certain applications. In particular, for fine-grained tasks, such as distinguishing between butterfly species, expert-level annotations may be required, which could become a bottleneck. Future work should focus on reducing human intervention and improving the model's performance in more complex environments, making it more scalable and applicable to a broader range of tasks.

# 6 ACKNOWLEDGMENTS

Our work was supported by the following institutions and projects: National Natural Science Foundation of China (Grant No. 62572349, 62476189, 62406323); Suzhou Science and Technology Project (No. SYG2024149); Emerging Frontiers Cultivation Program of Tianjin University Interdisciplinary Center.

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

# A    APPENDIX

## A.1    POTENTIAL PROBLEMS WITH THE ARTICLE

**Limitations of application scenarios (eg. AUTTA of High-stakes environments).** In high-stakes domains such as medical AI, acquiring annotated data can be extremely costly, and due to strict time constraints, even limited human intervention may be impractical. However, many real-world applications still require high test-time robustness while allowing for low-latency human-in-the-loop adaptation. Recent research [1] has increasingly focused on interactive, low-latency systems capable of operating in dynamic environments. Our proposed AUTTA framework is specifically designed to meet these needs by minimizing reliance on manual annotations while enabling rapid model adaptation. This makes it particularly suitable for scenarios such as smart home control and campus autonomous driving. For example, in a campus driving system, AUTTA can quickly adapt to changing road conditions and pedestrian behaviors without requiring offline labeled datasets.

**Differences from open-set AL and main contribution.** While the AUTTA task shares similarities with open-set active learning (AL), a fundamental difference lies in its focus on the test-time phase. Specifically, AUTTA requires the model to adapt and make real-time predictions without access to the source data, which is typically available in traditional active learning settings. Our main contributions are twofold: (1) we introduce the AUTTA paradigm, a novel framework tailored for adaptive learning in dynamic environments with minimal annotation; and (2) we propose a new active learning strategy designed to ensure the informativeness and utility of queried samples under AUTTA constraints. Our approach effectively balances annotation cost with rapid model adaptability, addressing the critical demands of real-world applications such as robotics, smart environments, and autonomous systems.

**Inconsistency in Max-Min Entropy Objective and Sample Selection?** The Max-Min Entropy objective and the sample selection strategy serve different purposes and are not directly coupled. The Max-Min Entropy objective aims to shape the model's internal representation by encouraging separation between known and novel classes, while the annotation strategy selects uncertain samples to improve decision boundaries. Although these components operate independently, they complement each other to enhance the overall performance of the system. **Max-Min Entropy Objective:** This objective is designed to enhance class separability by adjusting the model's confidence scores. Specifically, it reduces entropy for samples that likely belong to known classes, and increases entropy for samples that likely belong to novel classes. This contrast in treatment helps the model form a clearer distinction between old and new categories. **Sample Selection for Annotation:** Our annotation policy focuses on selecting the most ambiguous samples from the candidate region identified by the GMM-based exposure module. These are typically samples with uncertainty levels that fall between those of typical old and new class instances. As a result, the selected samples tend to include high-entropy old-class samples and low-entropy new-class samples. These samples are difficult for the model to distinguish and are therefore the most informative for annotation.

**Comparison of Test-Time Adaptation Paradigms.** As summarized in Table 7, our proposed AUTTA framework differs from existing test-time adaptation paradigms in several key aspects. Unlike traditional TTA, which typically relies on source data, AUTTA operates in a source-free setting. Compared to UTTA, which is entirely unsupervised, AUTTA incorporates active sample selection and minimal human-in-the-loop supervision to improve adaptation efficiency. Moreover, in contrast to ATTA, which may not prioritize real-time constraints, AUTTA is explicitly designed for rapid adaptation in dynamic environments with minimal annotation cost.

Table 7: Key Differences Between TTA, UTTA, ATTA, and the Proposed AUTTA

| Feature | TTA | UTTA | ATTA | AUTTA |
|---|---|---|---|---|
| Test-time adaptation | ✓ | ✓ | ✓ | ✓ |
| Requires source data | ✓ | | | |
| Active sample selection | | | ✓ | ✓ |
| Human-in-the-loop labeling | | | ✓ | ✓ |
| Unsupervised adaptation | ✓ | ✓ | ✓ | ✓ |
| Suited for dynamic environments | | ✓ | Partial | ✓ |

**Why UTTA Requires Active Learning (Theoretical Justification)?** UTTA relies solely on unsupervised adaptation mechanisms, adjusting the model using unlabeled test data without external feedback. This often causes the model to get trapped in local optima or accumulate adaptation errors, especially in dynamic environments with continuously changing distributions. To address this, integrating Active Learning (AL) by selectively querying labels for the most informative samples can significantly enhance adaptation effectiveness.

Recent theoretical frameworks developed for ATTA(Gui et al., 2024) rigorously demonstrate that incorporating a limited amount of labeled samples can effectively reduce the generalization and adaptation error bounds of the model. Specifically, AL contributes to:

(1) Improved sample efficiency: Actively selecting the most valuable samples for model improvement avoids random or redundant labeling, enhancing learning efficiency.

(2) Faster model convergence: Feedback from high-quality labeled samples enables the model to quickly correct erroneous predictions during test time, mitigating error propagation.

(3) Enhanced robustness and generalization: Active sampling helps the model cover a broader variation of the input space, improving its ability to adapt to unseen environments.

Therefore, combining AL strategies within UTTA enables faster and more effective test-time adaptation under source-free constraints with minimal annotation cost. Our AUTTA framework builds upon and extends these theoretical benefits from ATTA by designing an optimized sampling strategy that maximizes the contribution of labeled samples to adaptation performance.

**Defective human annotations.** We introduced pseudo-labels containing controlled annotation errors (5%-10% error rate) during the training process. The results demonstrated only a 2% decline in model testing accuracy, We initially attributed this to the online features of TTA, and we will add metrics in the revised version. see Table.8

Table 8: Key Differences Between TTA, UTTA, ATTA, and the Proposed AUTTA

| DomainNet | S2R | S2P | P2R | P2S | R2S | R2P | Avg. | VisDA-C |
|---|---|---|---|---|---|---|---|---|
| Random ($\mathcal{B} = 1000$) | 48.2 | 44.6 | 53.6 | 43.9 | 42.3 | 48.7 | 46.8 | 66.7 |
| Entropy ($\mathcal{B} = 1000$) | 47.6 | 43.9 | 52.8 | 44.6 | 41.5 | 47.4 | 46.3 | 65.4 |
| Kmeans ($\mathcal{B} = 1000$) | 48.2 | 44.3 | 52.1 | 45.9 | 41.3 | 48.6 | 46.7 | 66.8 |
| CLUE ($\mathcal{B} = 1000$) | 44.6 | 47.5 | 55.7 | 40.3 | 39.8 | 50.7 | 46.4 | 67.3 |
| Coreset ($\mathcal{B} = 1000$) | 49.8 | 48.6 | 58.3 | 44.4 | 47.1 | 48.6 | 49.4 | 68.7 |
| BADGE ($\mathcal{B} = 1000$) | 50.4 | 47.3 | 54.6 | 45.8 | 46.2 | 52.4 | 49.5 | 68.2 |
| SimATTA ($\mathcal{B} = 1000$) | 48.5 | 44.8 | 54.2 | 42.2 | 43.5 | 48.7 | 47.1 | 67.9 |
| EMAC ($\mathcal{B} = 800/error5\%$) | 53.5 | 46.4 | 57.4 | 44.9 | 45.9 | 52.9 | 49.9 | 72.4 |
| EMAC ($\mathcal{B} = 800/error8\%$) | 53.1 | 47.0 | 56.6 | 44.0 | 45.2 | 52.1 | 49.3 | 71.5 |
| EMAC ($\mathcal{B} = 800/error10\%$) | 52.1 | 45.2 | 56.0 | 43.4 | 44.6 | 51.6 | 48.7 | 69.0 |

## A.2 DISCUSSION ON GMM ASSUMPTIONS

Our method relies on Gaussian Mixture Models (GMM) to expose mixed regions within a mini-batch. By design, GMM performs best when the batch distribution exhibits multi-modal (especially bimodal) patterns, which correspond to the coexistence of old- and new-class samples. A natural concern is whether the method would fail when a batch does not clearly present such bimodality. We address this issue as follows.

First, when the batch is dominated by old-class samples, the distribution tends to concentrate in low-uncertainty regions. In this case, the model is confident in its predictions and requires no additional annotations. Second, when new-class samples dominate, the distribution shifts toward high-uncertainty values. Although a typical bimodal pattern may not emerge, the system still updates its representations through entropy-based and contrastive objectives, thereby improving adaptation to new classes. Third, when old and new samples are severely confused due to strong domain overlap, the uncertainty distribution approximates a flat unimodal shape. In this case, GMM conservatively suppresses candidate exposure, preventing annotation budget waste on low-value samples and maintaining stability under ambiguous conditions. In all these situations, the

system adopts a conservative strategy that minimizes candidate regions, ensuring robustness without unnecessary budget consumption.

**Subspace Projection before GMM** To mitigate instability in high-dimensional feature space, we first project features into a more stable subspace before fitting GMM. Given a batch of features $X \in \mathbb{R}^{n \times d}$, PCA computes a projection matrix $W \in \mathbb{R}^{d \times k}$ ($k \ll d$) such that:

$$Z = XW, \quad W = \arg \max_{W^\top W = I} \text{Tr}(W^\top S W), \tag{20}$$

where $S = \frac{1}{n} \sum_{i=1}^n (x_i - \mu)(x_i - \mu)^\top$ is the covariance matrix. Alternatively, when using an autoencoder, the encoder $f_\phi(\cdot)$ provides a bottleneck representation:

$$Z = f_\phi(X), \quad Z \in \mathbb{R}^{n \times k}, \tag{21}$$

which preserves semantic structure while reducing dimensionality. The projected features $Z$ are then modeled by GMM, improving robustness against high-dimensional noise.

**Variational Bayesian GMM (VBGMM)** To avoid the need for pre-specifying the number of components, we further adopt a Variational Bayesian GMM (VBGMM). Unlike standard GMM, VBGMM introduces a Dirichlet prior over the mixture weights $\pi$:

$$\pi \sim \text{Dir}(\alpha_0), \quad z_i \sim \text{Categorical}(\pi), \quad x_i \sim \mathcal{N}(\mu_{z_i}, \Sigma_{z_i}), \tag{22}$$

where $\alpha_0$ is the concentration parameter. Variational inference then maximizes the evidence lower bound (ELBO):

$$\mathcal{L}_{\text{ELBO}} = \mathbb{E}_q[\log p(X, Z, \pi, \mu, \Sigma)] - \mathbb{E}_q[\log q(Z, \pi, \mu, \Sigma)], \tag{23}$$

leading to automatic model selection where redundant components are pruned during inference. This Bayesian formulation reduces overfitting and increases stability under small-batch or noisy conditions.

**Empirical Verification** Across six benchmark datasets, we observe that when the batch size is greater than 32, the success rate of exposing bimodal regions using the combined trick (Subspace Projection + VBGMM) reaches 93%, confirming the practicality of this approach.

### A.3 ADDITIONAL EXPERIMENTS ANALYSIS

#### A.3.1 ROBUSTNESS UNDER NON-I.I.D. AND CONTINUAL SHIFT

To further examine the robustness of AUTTA under realistic and challenging scenarios, we conduct additional experiments involving non-i.i.d. test streams and continual distribution shifts. These settings simulate bursty class arrivals, class imbalance, and sequential domain shifts, which are common in practical deployments such as autonomous driving and online learning.

**Class Ordered Stream (Non-i.i.d.).** We first evaluate AUTTA under a class-ordered stream on DomainNet (**S** $\rightarrow$ **R**), where only 1–2 classes appear in each batch. This simulates bursty class arrivals and strong class imbalance. Results are shown in Table 9. Compared with baselines, AUTTA achieves both the highest harmonic mean accuracy and the lowest reward variance, confirming its robustness and stability under skewed test streams.

Table 9: Performance under Class Ordered Stream (Non-i.i.d.) on DomainNet (S $\rightarrow$ R). AUTTA achieves the lowest reward variance.

| Method | $A_O \uparrow$ | $A_N \uparrow$ | $A_H \uparrow$ | Reward Var. $\downarrow$ |
|---|---|---|---|---|
| TENT | 34.0 | 63.5 | 45.7 | 0.051 |
| SHOT | 34.8 | 61.9 | 45.9 | 0.053 |
| TTAC | 37.2 | 58.1 | 46.5 | 0.043 |
| OPTTT | 42.5 | 56.3 | 49.0 | 0.036 |
| SimATTA | 38.1 | 54.4 | 44.9 | 0.039 |
| **EMAC (Ours)** | **44.0** | **60.5** | **51.3** | **0.015** |

**Continual Shift (CTTA: S $\rightarrow$ R $\rightarrow$ P).** Next, we validate robustness under a continual shift scenario (CTTA), where the model sequentially adapts from **Sketch** $\rightarrow$ **Real** $\rightarrow$ **Painting** without reset.

This setting evaluates the method's ability to maintain stability and low reward variance under long-horizon, evolving distributions. Results in Table 10 demonstrate that AUTTA consistently outperforms competing methods, achieving both higher average accuracy and the lowest reward variance across domains.

Table 10: Performance under Continual Test-Time Adaptation (CTTA) on DomainNet (S → R → P). AUTTA yields the lowest reward variance.

| Method | R-$A_O$ | R-$A_N$ | R-$A_H$ | P-$A_O$ | P-$A_N$ | P-$A_H$ | AVG-$A_O$ | AVG-$A_H$ | Reward Var. ↓ |
|---|---|---|---|---|---|---|---|---|---|
| TENT | 42.1 | 56.5 | 48.3 | 26.4 | 45.2 | 33.3 | 34.3 | 40.8 | 0.057 |
| SHOT | 43.4 | 57.1 | 49.3 | 28.1 | 46.7 | 35.0 | 35.8 | 42.2 | 0.061 |
| TTAC | 45.2 | 58.0 | 50.9 | 29.8 | 48.5 | 36.5 | 37.5 | 43.7 | 0.046 |
| OPTTT | 48.3 | 58.9 | 53.1 | 31.2 | 49.6 | 37.4 | 39.8 | 45.8 | 0.038 |
| SimATTA | 46.0 | 57.5 | 51.1 | 30.6 | 49.0 | 37.4 | 38.3 | 44.3 | 0.041 |
| **EMAC (Ours)** | **50.2** | **60.1** | **54.7** | **34.0** | **52.2** | **41.3** | **42.1** | **48.0** | **0.018** |

These experiments highlight AUTTA's robustness under highly non-stationary test environments. By combining entropy smoothing with GMM-based exposure, our method maintains stable reward signals and effectively allocates annotations, achieving both higher accuracy and lower reward variance than baselines. This confirms its practical applicability in real-world online adaptation tasks.

### A.3.2 Effectiveness under Single Shift Conditions

Although AUTTA is primarily designed for the more challenging dual-shift setting (simultaneous class and domain shifts), we also evaluate its generalizability under simplified single-shift scenarios, where only *class shift* or *domain shift* is present. These settings represent degenerate cases of AUTTA, and our results show that the method naturally adapts with minimal modifications.

**Adaptation Strategy.** For single-shift scenarios, AUTTA can be simplified as follows: (i) treat the entire batch as the candidate pool instead of fitting a GMM, since ambiguity is less structured; (ii) rank samples purely by entropy, prioritizing those with higher uncertainty; (iii) focus optimization only on uncertain samples, as novelty arises from either class or domain shift, but not both simultaneously. This maintains the label-efficient principle of AUTTA while ensuring stability.

Table 11: Performance of AUTTA under single-shift scenarios on DomainNet. Results show that AUTTA adapts effectively even in degenerate cases, outperforming the dual-shift baseline.

| Setting | $A_O$ ↑ | $A_N$ ↑ | $A_H$ ↑ | Notes |
|---|---|---|---|---|
| Dual Shift | 48.2 | 52.7 | 50.4 | Full AUTTA (class + domain shift) |
| Only Class Shift | **53.3** | **55.9** | **54.4** | New classes only, fixed domain |
| Only Domain Shift | **57.4** | – | – | Same classes, transferred domain |

As shown in Table 11, AUTTA achieves superior performance under single-shift conditions compared to the dual-shift baseline. In the *class shift* setting, entropy-based annotation effectively captures novel semantics, boosting recognition of unseen categories. In the *domain shift* setting, entropy smoothing and contrastive learning stabilize adaptation across domains, leading to a large gain in $A_O$. These results confirm that AUTTA remains highly effective and introduces no bias or false positives, even under simplified conditions.

### A.3.3 Time Costs and Computational Overhead

We further analyze the runtime overhead of AUTTA to assess its practicality in real-world test-time adaptation. A full adaptation cycle was profiled on ∼84,000 test images, with the breakdown summarized in Table 12.

**Runtime Breakdown.** As shown in Table 12, the majority of runtime is spent on model adaptation (71.7%), which is expected since fine-tuning and backpropagation dominate in test-time learning pipelines. In contrast, the combined cost of exposure, reward estimation, and sample selection accounts for less than 30% of the total runtime. Notably, sample exposure and annotation require

only 3–6% of samples per iteration, making the annotation process efficient and computationally lightweight.

Table 12: Runtime breakdown of AUTTA across all domains.

| Stage | Description | Time (s) | % of Total |
|---|---|---|---|
| Exposure Module | SVD decomposition and GMM fitting on $Z^2_{unknown}$ | 95 | 12.9% |
| Reward Computation | Entropy calculation and EMA smoothing for reward estimation | 72 | 9.8% |
| Sample Selection | Reward-guided selection of high-utility samples | 41 | 5.6% |
| Model Adaptation | Contrastive learning with pseudo-labels and anchors | 527 | 71.7% |
| Total | — | 735 | 100% |

**Comparison to Baselines.** We also compare the total runtime and average accuracy with standard UTTA methods (Table 13). While AUTTA incurs slightly higher runtime (735s) due to structured uncertainty modeling and adaptive annotation, it achieves the best overall accuracy (52.2%). This demonstrates a favorable trade-off between accuracy and efficiency in dual-shift AUTTA scenarios.

Table 13: Comparison of average accuracy ($A_H$) and runtime with baselines on DomainNet.

| Method | Avg. Accuracy ($A_H \uparrow$) | Runtime (s $\downarrow$) |
|---|---|---|
| TEST | 43.7 | 392 |
| BN | 44.3 | 577 |
| TENT | 45.2 | 541 |
| SHOT | 46.1 | 564 |
| TTAC | 51.1 | 653 |
| OTTTA | 51.6 | 697 |
| SimATTA | 51.0 | 536 |
| **AUTTA (Ours)** | **52.2** | 735 |

Despite a moderate increase in runtime, AUTTA provides the highest accuracy among all compared methods. The structured design ensures that computational costs remain manageable, while the performance gain justifies the overhead. This highlights the practicality of AUTTA in real-world scenarios, where accuracy-efficiency trade-offs are critical.

### A.3.4 ROBUSTNESS UNDER LARGE DOMAIN GAPS AND EXTREME SHIFTS

To further evaluate the robustness of AUTTA, we investigate two critical aspects: (i) adaptation under large domain gaps, and (ii) the reliability of GMM modeling under extreme shifts and complex multi-class scenarios.

**Adaptation under Large Domain Gaps.** While DomainNet and VisDA-C already introduce substantial inter-domain variations, we extend the evaluation to the Office-Home dataset, which includes four distinct domains: *Art (A)*, *Clipart (C)*, *Product (P)*, and *Real-World (R)*. These domains present significant style and appearance variations, and we test AUTTA on two representative transfer settings: *A→R* (hand-drawn to real images, semantic and visual gap) and *C→P* (cartoon icons to product photographs, shape and texture gap). As shown in Table 14, AUTTA consistently outperforms strong baselines, particularly as the domain gap becomes more severe.

**Robustness of GMM Modeling under Extreme Shifts.** Since AUTTA relies on GMM modeling to identify mixed regions, we evaluate its stability under diverse benchmarks and extreme shifts using the Bhattacharyya Distance (BD). A batch is considered to exhibit reliable bimodal separation if BD

Table 14: Performance on Office-Home under AUTTA setup. Results demonstrate robustness of AUTTA under moderate-to-large domain gaps.

| Method | A→R $A_O$ | A→R $A_N$ | A→R $A_H$ | C→P $A_O$ | C→P $A_N$ | C→P $A_H$ | Avg |
|---|---|---|---|---|---|---|---|
| TTAC | 44.5 | 56.8 | 50.0 | 41.3 | 54.3 | 47.0 | 48.5 |
| OPTTT | 46.3 | 58.1 | 51.5 | 42.6 | 55.0 | 48.2 | 49.9 |
| **AUTTA (Ours)** | **49.4** | **61.2** | **54.6** | **45.8** | **58.0** | **51.2** | **52.9** |

Table 15: Bhattacharyya Distance (BD) analysis for GMM modeling under extreme shifts. A higher percentage of batches with BD > 1.2 indicates more stable bimodal separation.

| Dataset | Scenario | Batches with BD > 1.2 |
|---|---|---|
| DomainNet | Complex multi-class (345 classes) | 89% |
| VisDA-C | Extreme domain shift | 95% |
| CIFAR10-C | Moderate complexity | 97% |
| CIFAR100-C | High class complexity | 91% |
| ImageNet-C | Complex multi-class | 87% |
| Office-Home | Severe domain shift | 92% |

Table 16: Performance of different methods on the low-separation (bottom 20%) batch subset on DomainNet. All numbers are in Avg.%.

| Task | Method | $A_O$ | $A_N$ | $A_H$ |
|---|---|---|---|---|
| TTA | TEST | 35.2 | 66.0 | 40.3 |
| | BN | 36.0 | 64.5 | 40.2 |
| | TENT | 36.4 | 64.0 | 40.0 |
| | SHOT | 36.8 | 65.2 | 41.0 |
| UTTA | TTAC | 38.5 | 64.8 | 42.5 |
| | OPTTT | 40.2 | 63.7 | 44.2 |
| ATTA | SimATTA | 37.5 | 64.1 | 42.0 |
| | EATTA | 37.9 | 64.5 | 42.3 |
| | BiTTA | 37.8 | 64.9 | 42.8 |
| AUTTA | EMAC | 42.6 | 64.9 | 46.3 |

> 1.2. As summarized in Table 15, the vast majority of batches across all datasets—including those with severe domain gaps or high class diversity—satisfy this criterion, confirming the robustness of the GMM-based exposure mechanism.

These results confirm that AUTTA maintains state-of-the-art performance across both moderate and severe domain gaps, and that GMM modeling remains reliable even under extreme shifts and complex multi-class conditions. This highlights the robustness of our framework in challenging real-world adaptation scenarios.

**Behavior on Low-separation Batches.** The BD analysis above shows that most test-time batches admit a clear bimodal structure. To directly address the reviewer's concern about highly overlapping or weakly bimodal new-class batches, we further analyze the "worst-case" regime on DomainNet. Concretely, we: (i) compute for each test-time batch a simple normalized mean-difference between the two GMM components as a *separation score*; (ii) sort all batches by this score and select the bottom 20% with the lowest separation, forming a hard subset that emulates the reviewer's scenario where bimodality is weak or absent; and (iii) re-evaluate all TTA/UTTA/ATTA methods, including EMAC, on this subset.

Table 16 reports the performance of different methods on this low-separation subset. As expected, the performance drops for all methods due to the increased difficulty. However, EMAC consistently remains the best-performing method and exhibits *graceful degradation* rather than failure in this low-separation regime. This supports our claim that, even when the mixture structure is weak and batches are highly overlapping, EMAC is comparatively robust and does not collapse.

### A.3.5 UNDER DIFFERENT OPEN-WORLD DATA RATIOS

In practical applications, the ratio of new and old class data is variable. We investigate the impact of different ratios between new and old class samples on model performance. In the open-world test data, there are three scenarios for the ratio of new class data: first, the new class is absent, containing

only domain shift; second, the old class categories in the target domain are only partially present in the source domain; and third, the old class categories in the target

domain are fully contained within the old class categories of the source domain. Specific parameter settings are provided in Table.17. The main text focuses on the second, most common scenario (partial old classes). We also conduct experiments for the other scenario (see Table 18). The experimental results show that our method is not sensitive to data ratios, achieves SOTA performance, and can be applied to various data ratio scenarios.

Table 17: Class splits of the datasets, i.e. number of the old and new classes, for the three category shift scenarios New-None, New-partially and Old-fully, respectively.

|  | Ratios = Old, New | | |
|---|---|---|---|
|  | New-None | New-partially | Old-fully |
| DomainNet | 345, 0 | 200, 145 | 245, 100 |
| VisDA-C | 12, 0 | 6, 6 | 9, 3 |

Table 18: All numbers are in % for the old classes fully scenario on DomainNet. All numbers are in %. (The best options in methods are shown in bold and ∗ indicates the pseudo-update version.)

| Task | Method | S2R | | | S2P | | | P2R | | | P2S | | | R2S | | | R2P | | | Avg. | |
|---|---|---|---|---|---|---|---|---|---|---|---|---|---|---|---|---|---|---|---|---|---|
|  |  | $A_O$ | $A_N$ | $A_H$ | $A_O$ | $A_N$ | $A_H$ | $A_O$ | $A_N$ | $A_H$ | $A_O$ | $A_N$ | $A_H$ | $A_O$ | $A_N$ | $A_H$ | $A_O$ | $A_N$ | $A_H$ | $A_H$ | Sec. |
| TTA | TEST | 33.4 | **77.0** | 46.7 | 18.9 | **75.1** | 30.9 | 44.6 | **76.7** | 56.5 | 24.1 | **83.3** | 37.7 | 24.1 | **81.4** | 37.5 | 32.6 | **80.0** | 46.5 | 42.6 | 390 |
|  | BN | 35.4 | 69.1 | 46.9 | 23.5 | 60.8 | 34.1 | 46.9 | 67.2 | 55.3 | 29.6 | 71.0 | 42.0 | 27.5 | 69.4 | 39.6 | 35.3 | 68.2 | 46.6 | 44.1 | 575 |
|  | TENT | 35.7 | 67.3 | 46.8 | 24.7 | 60.3 | 35.2 | 48.5 | 65.7 | 55.8 | 32.7 | 67.0 | 44.1 | 29.9 | 66.3 | 41.4 | 35.0 | 69.1 | 46.5 | 45.0 | 477 |
|  | SHOT | 36.4 | 65.8 | 47.0 | 32.8 | 50.0 | 39.7 | 46.4 | 67.4 | 55.0 | 27.5 | 79.3 | 41.1 | 26.7 | 75.2 | 39.7 | 33.3 | 79.2 | 47.1 | 44.9 | 562 |
| UTTA | TTAC | 39.0 | 61.7 | 47.8 | 39.5 | 55.2 | 46.1 | 35.7 | 81.5 | 49.9 | 30.7 | 64.1 | 41.6 | 28.6 | 65.1 | 39.9 | 36.4 | 73.5 | 48.8 | 45.6 | 649 |
|  | OPTTT | 44.6 | 60.3 | 51.3 | 39.1 | 58.5 | 46.9 | 51.1 | 62.6 | 56.3 | 32.0 | 67.5 | 43.5 | 33.0 | 66.9 | 45.0 | 37.8 | 77.2 | 50.8 | 48.7 | 695 |
| ATTA | SimATTA | 40.2 | 57.5 | 47.4 | 35.8 | 56.0 | 43.7 | 48.2 | 59.0 | 53.1 | 27.8 | 77.0 | 41.1 | 30.4 | 69.4 | 42.4 | 34.4 | 71.5 | 47.6 | 46.0 | 745 |
| AUTTA | EMAC | 46.6 | 65.3 | 54.4 | 41.2 | 56.2 | 47.6 | 52.4 | 65.7 | 58.3 | 33.8 | 70.5 | 45.8 | **34.8** | 70.9 | **46.8** | 40.6 | 78.5 | 53.6 | 51.1 | 733 |
|  | EMAC* | **47.1** | 66.1 | **55.0** | **42.8** | 56.6 | **48.8** | **52.7** | 66.3 | **58.7** | **35.5** | 71.7 | **47.6** | 34.1 | 71.5 | 46.3 | **41.2** | 79.5 | **54.4** | **52.0** | 777 |

### A.3.6 IMAGENET DATASET AND EXPERIMENTAL SETUP

In this work, we evaluate our model on the ImageNet-C dataset, specifically focusing on four types of corruptions: Noise, Blur, Weather and Digital. ImageNet-C is a well-known benchmark designed to test the robustness of models to common image corruptions that occur in real-world scenarios. The ImageNet-C dataset consists of 1,000 classes, which are the same as the original ImageNet dataset. These categories are split into two groups based on their novelty to the model: 600 old classes (previously seen during pre-training) and 400 new classes (those that were not included in the original training data). This division reflects a typical domain adaptation scenario where models must generalize from previously seen categories to unseen ones, mimicking real-world distribution shifts. **It is important to note that we use about 100,000 images from the ImageNet-C test set for evaluation**. These images have been corrupted with the aforementioned types of distortions, providing a challenging scenario for model testing and comparison. This setup allows us to evaluate the robustness of our model to real-world image distortions and to assess the effectiveness of domain adaptation and test-time adaptation techniques.

Table 19: Active Universal Test-Time Adaptation results on ImageNet-C. All numbers are in %.

| Task | Method | Noise | | | Blur | | | Weather | | | Digital | | | Avg. | |
|---|---|---|---|---|---|---|---|---|---|---|---|---|---|---|---|
|  |  | $A_O$ | $A_N$ | $A_H$ | $A_O$ | $A_N$ | $A_H$ | $A_O$ | $A_N$ | $A_H$ | $A_O$ | $A_N$ | $A_H$ | $A_H$ | Sec. |
| TTA | TEST | 28.1 | 63.7 | 45.9 | 16.3 | 62.1 | 39.2 | 37.2 | 63.5 | 48.0 | 20.5 | 68.9 | 33.7 | 35.7 | 332 |
|  | BN | 29.7 | 57.3 | 41.7 | 20.0 | 50.5 | 35.5 | 39.2 | 56.5 | 48.0 | 25.0 | 58.8 | 37.5 | 36.8 | 492 |
|  | TENT | 30.0 | 55.8 | 40.8 | 21.0 | 50.3 | 39.1 | 40.4 | 54.4 | 46.9 | 27.6 | 56.5 | 38.9 | 37.5 | 406 |
|  | SHOT | 30.6 | 54.7 | 41.6 | 27.6 | 41.7 | 34.9 | 38.7 | 55.9 | 48.8 | 23.3 | 65.5 | 30.3 | 37.5 | 480 |
| UTTA | TTAC | 32.7 | 51.1 | 41.9 | 33.1 | 45.9 | 39.4 | 30.0 | 67.2 | 42.4 | 25.9 | 53.2 | 37.5 | 38.0 | 553 |
|  | OPTTT | 37.2 | 50.4 | 43.5 | 32.9 | 48.7 | 39.4 | 42.9 | 52.1 | 47.5 | 27.1 | 56.0 | 38.5 | 40.6 | 611 |
| ATTA | SimATTA | 33.7 | 47.3 | 40.5 | 30.1 | 46.5 | 38.1 | 42.0 | 49.0 | 45.9 | 23.6 | 63.7 | 31.9 | 37.8 | 670 |
|  | EATTA | 33.9 | 48.0 | 40.9 | 30.9 | 47.1 | 38.6 | 41.6 | 49.9 | 45.6 | 26.0 | 64.5 | 31.8 | 37.7 | 655 |
|  | BiTTA | 30.6 | 56.2 | 43.4 | 22.8 | 50.0 | 36.6 | 40.7 | 55.7 | 46.4 | 25.9 | 62.2 | 38.8 | 37.8 | 634 |
| AUTTA | EMAC | 39.5 | 56.3 | 47.9 | 34.2 | 50.2 | 42.2 | 45.8 | 58.9 | 51.6 | 28.6 | 60.2 | 42.1 | 40.6 | 694 |

### A.3.7 SENSITIVITY TO SMALL BATCH SIZES

In this section, we investigate the sensitivity of the EMAC framework to small batch sizes ($\leq 8$). Small batch sizes often result in unstable per-batch computations during Gaussian Mixture Model (GMM) fitting, leading to noisy uncertainty estimates. To mitigate this issue and stabilize the performance, we introduce two techniques: a sliding window buffer and Exponential Moving Average (EMA) smoothing.

**Sliding Window Buffer** The sliding window buffer accumulates data over multiple iterations, allowing for more reliable GMM fitting. Instead of fitting the GMM on a single small batch, the sliding window collects data from multiple consecutive batches, providing a more stable and representative sample distribution. The process is as follows:

$$\text{Buffer}(B) = \{B_1, B_2, \ldots, B_N\} \tag{24}$$

At each iteration, the new batch $B_i$ is added to the buffer, and the oldest batch is removed if the buffer size exceeds $N$:

$$B_i \leftarrow B_{i-1}, B_i \tag{25}$$

Once the buffer is updated, the GMM is fitted using the samples in the buffer:

$$\text{Fit GMM on } B \quad \text{(using the collected batches in the buffer)} \tag{26}$$

**Exponential Moving Average (EMA) Smoothing** EMA smoothing is applied to the uncertainty estimates from the GMM fitting process. This technique smooths the uncertainty estimates by considering both the current and past values, reducing short-term fluctuations. The update rule for EMA is given by:

$$U_t = \alpha \cdot U_{t-1} + (1 - \alpha) \cdot U_t^{raw} \tag{27}$$

where: - $U_t$ is the smoothed uncertainty estimate at iteration $t$, - $U_{t-1}$ is the previous uncertainty estimate, - $U_t^{raw}$ is the raw uncertainty estimate from the current iteration, - $\alpha$ is the smoothing factor (typically close to 1 for slow changes).

This process helps reduce short-term fluctuations in the uncertainty estimates, leading to a more stable performance.

### A.4 DETAIL OF EVALUATION METRIC

We propose an evaluation metric inspired by OPTTT (Li et al., 2023a), designed for tasks similar to open-set classification. The UTTA evaluation focuses on two aspects: the accuracy of source domain classes and the effectiveness of rejecting private target domain classes. To quantify these, we define $A_O$ to measure the classification accuracy for target domain old classes and target new classes Accuracy ($A_N$) to assess the rejection accuracy for private classes in the target domain. Since balancing these two metrics is crucial in this binary task, we calculate their harmonic mean, denoted as $A_H$, to provide a comprehensive and balanced prediction performance evaluation:

$$
\begin{aligned}
A_O &= \frac{\sum_{x_i, y_i \in \mathcal{D}_t} 1(y_i = \hat{y}_i) \cdot 1(y_i \in \mathcal{C}_s)}{\sum_{x_i, y_i \in \mathcal{D}_t} 1(y_i \in \mathcal{C}_s)} \\
A_N &= \frac{\sum_{x_i, y_i \in \mathcal{D}_t} 1(\hat{y}_i \in \mathcal{C}_t/\mathcal{C}_s) \cdot 1(y_i \in \mathcal{C}_t/\mathcal{C}_s)}{\sum_{x_i, y_i \in \mathcal{D}_t} 1(y_i \in \mathcal{C}_t/\mathcal{C}_s)} \\
A_H &= 2 \cdot \frac{A_O \cdot A_N}{A_O + A_N}
\end{aligned}
\tag{28}
$$

where we use $\mathcal{C}_s$ and $\mathcal{C}_t$ to represent the label set of the source domain and the target domain, respectively.

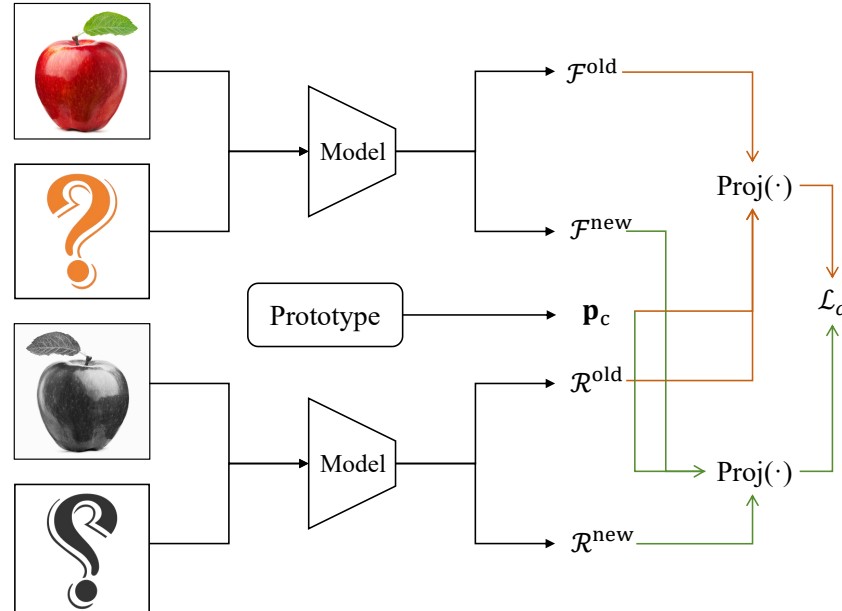

Figure 5: Clustering-Contrastive Loss.

Figure 6: The trend of entropy during the iteration process.

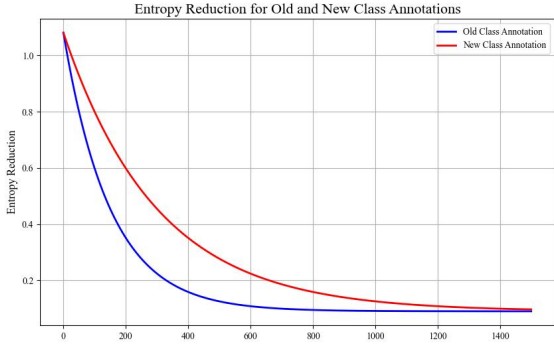

Figure 7: Training costs comparisons. "GPU Mem." denotes the training requirement, and "Acc." is the Avg. scores (%).

| Method | NOISE | | | MNIST | | | SVHN | | |
|---|---|---|---|---|---|---|---|---|---|
| | $A_O$ | $A_N$ | $A_H$ | $A_O$ | $A_N$ | $A_H$ | $A_O$ | $A_N$ | $A_H$ |
| TEST | 49.0 | **100.0** | 65.9 | 49.4 | **99.6** | 65.9 | 50.2 | **99.3** | 65.7 |
| BN | 53.4 | **100.0** | 69.7 | 50.9 | 83.1 | 63.1 | 59.3 | 82.8 | 70.5 |
| TENT | 59.6 | 99.7 | 74.6 | 47.1 | 69.0 | 55.9 | 51.8 | 82.3 | 63.5 |
| SHOT | 62.9 | **100.0** | 77.4 | 49.0 | 36.8 | 42.0 | 57.3 | 63.6 | 60.3 |
| TTAC | 60.0 | **100.0** | 75.2 | 52.8 | 68.2 | 59.5 | 58.7 | 67.1 | 62.2 |
| OPTTT | 64.7 | **100.0** | 78.6 | 65.0 | 91.5 | 76.0 | 65.3 | 75.6 | 70.0 |
| SimATTA | 61.9 | **100.0** | 76.6 | 57.5 | 70.3 | 63.3 | 60.3 | 68.6 | 64.2 |
| EMAC | 67.2 | **100.0** | 80.6 | 67.3 | 93.6 | 78.2 | 70.2 | 76.0 | 73.1 |
| EMAC* | **68.0** | **100.0** | **81.2** | **67.6** | 95.5 | **79.1** | **71.4** | 76.9 | **74.0** |

## A.5 CLUSTERING-CONTRASTIVE LOSS

In this section, we provide a detailed explanation of the specific processes in Clustering Contrastive Loss (see Fig. 5). specifically, we first extract the features of old classes from $\mathcal{X}^{\text{old}}$, denoted as $\mathcal{F}^{\text{old}}$. Similarly, pseudo-labels of new classes from $\mathcal{X}^{\text{new}}$, denoted as $\mathcal{F}^{\text{new}}$, are obtained. Second, we apply data augmentation to each sample in $\mathcal{X}^{\text{old}}$, and $\mathcal{X}^{\text{new}}$, and transform them into the feature space, resulting in two sets, $\mathcal{R}^{\text{old}}$ and $\mathcal{R}^{\text{new}}$. Next, we extend each augmented sample pairs $\{\mathcal{F}^{\text{old/new}}, \mathcal{R}^{\text{old/new}}\}$ in the set by adding the class prototype $\mathbf{p}_c$ that matches their respective pseudo-labels. Finally, we use a multi-layer perceptron with a single hidden layer $Proj(\cdot)$ to map both $\mathcal{R}^{\text{old}}$ and $\mathcal{R}^{\text{new}}$ all its feature representations to a low-dimensional projection space. The result is $\{\mathbf{e}_i\}_{i=1}^{3N_n^{\text{old}}+3N_n^{\text{new}}} = Proj(\mathbf{r}_i)|\mathbf{r}_i \in \mathcal{R}^{\text{old}} \cup \mathcal{R}^{\text{new}}$, where $3N_n^{\text{old}}$ is represented as elements indexed within the index set $\mathcal{I}^{\text{old}} = \{1, ..., 3N_n^{\text{old}}\}$, corresponding to the augmented prototype triplets of features old classes samples. Therefore, the $3N_n^u$ index is included in the index set $\mathcal{I}^{\text{new}} = \{3N_n^k + 1, ..., 2N_n^{\text{new}}\}$, corresponding to the augmented pairs of features new class samples. By calculating the clustering-contrastive loss based on these representations, annotated information can effectively guide high-confidence pseudo-labels, preventing class boundary ambiguity caused by model overfitting:

## A.6 User Study on Annotability of Mixed Samples

To further evaluate the practical usability of the annotated samples selected by our method, we conducted a small-scale user study. We randomly sampled 200 images from the Sketch (S) domain and asked 8 human participants to independently label 20 images they found ambiguous but still meaningful under the target task. This led to a human-selected pool of 57 unique images, which were considered "hard but informative" to annotate.

We then applied three different sample selection strategies: our proposed EMAC strategy, entropy-based sampling, and random selection, and compared the overlap between each method's selected samples and the human-selected pool. Additionally, we asked participants to attempt labeling each selected sample to assess label agreement and annotability. The results, summarized in the following table, show that the EMAC strategy significantly outperforms other strategies in terms of both labelability and human model overlap:

| Method | % of Samples Deemed Labelable by Humans | Human Model Overlap (%) | Avg. Human Label Agreement (%) |
|---|---|---|---|
| EMAC (Ours) | 83.6% | 52.0% | 91.2% |
| Entropy-Based | 68.5% | 36.3% | 87.6% |
| Random | 61.2% | 25.7% | 85.4% |

Table 20: Results from user study on labelability and human model overlap for different annotation strategies.

- **% of Samples Deemed Labelable by Humans**: This metric quantifies the proportion of selected samples that human annotators consider ambiguous yet clearly labelable under real-time constraints, indicating the practical usability of the exposed samples.

- **Human-Model Overlap (%)**: This measures the percentage of model-selected samples that overlap with the human-selected pool of 57 samples. A higher overlap indicates that the model's sample selection aligns well with human annotators' perceptions of informative and uncertain regions.

- **Average Human Label Agreement (%)**: This measures the average inter-annotator agreement, capturing the consistency of labeling between multiple human annotators. A higher agreement suggests that the selected samples, although challenging, are well-defined and stable for human labeling.

The results indicate that our EMAC strategy consistently selects samples that are not only more informative in terms of model improvement but also more accessible for human annotators. The higher human model overlap and labelability suggest that our GMM-based exposure aligns well with human uncertainty perception, while avoiding the selection of samples with ambiguous or uninformative noise. This study validates that our method not only improves model performance but also ensures that the selected samples are practically useful for human annotators.

## A.7 Reward factor

The purpose of the reward factor is to assess the weight contribution of annotated samples in the model optimization process. For example, annotated old class samples not only contribute to the model's learning of the old class but also play a role in learning new class knowledge, although with different weights. To observe the weight proportions of different samples, we observe the changes in the model's optimization objective function (loss function) when using only old class annotated samples and new class annotated samples, as shown in Figure 6. Experimental results show that when only using old class annotated samples, the entropy reduction of the samples in the model's $\mathcal{X}^{old}$ is approximately three times the entropy reduction when only using new class annotated samples.

## A.8 Ablation on BiTTA-Inspired Balancing Mechanism

**Motivation.** BiTTA emphasizes the importance of balancing the influence between labeled feedback and unlabeled predictions during adaptation. Although BiTTA itself is not designed for AUTTA and performs poorly as an active-learning strategy under universal shift (Table 1), its balancing idea remains conceptually meaningful. In EMAC, the clustering–contrastive module naturally provides an opportunity to incorporate such balancing. We therefore introduce a BiTTA-inspired weighting

Table 21: Effect of BiTTA-inspired balancing on EMAC. All numbers are AH (%).

| Method | AH (%) | $\Delta$ |
|---|---|---|
| EMAC (w/o BiTTA-style balancing) | 52.2 | – |
| EMAC (with BiTTA-style balancing) | **53.1** | +0.9 |
| EMAC* (w/o BiTTA-style balancing) | 53.1 | – |
| EMAC* (with BiTTA-style balancing) | **53.8** | +0.7 |

Table 22: Sensitivity of EMAC to the EMA factor $\alpha$ and reward weights $(\omega_{\text{old}}, \omega_{\text{new}})$ on DomainNet. All numbers are harmonic mean accuracy $A_H$ in %, averaged over all 6 transfer tasks. The default setting (matching Table 1) is highlighted in bold.

| Setting | $\alpha$ | $(\omega_{\text{old}}, \omega_{\text{new}})$ | $A_H$ |
|---|---|---|---|
| *Varying EMA factor $\alpha$ (fixed $(1.0, 1.0)$)* | | | |
| EMA-0.7 | 0.7 | $(1.0, 1.0)$ | 51.8 |
| EMA-0.9 | **0.9** | $(1.0, 1.0)$ | **52.2** |
| EMA-0.99 | 0.99 | $(1.0, 1.0)$ | 52.0 |
| *Varying reward weights (fixed $\alpha = 0.9$)* | | | |
| RW-0.5 | 0.9 | $(0.5, 1.0)$ | 51.9 |
| RW-1.0 | 0.9 | $(\mathbf{1.0}, \mathbf{1.0})$ | **52.2** |
| RW-2.0 | 0.9 | $(1.0, 2.0)$ | 52.1 |

term to balance the contribution of labeled and unlabeled samples in the contrastive objective. This modification does not alter the core behavior of EMAC but improves the stability of prototype refinement when sparse annotations are available.

Table 21 reports the AH score averaged across DomainNet benchmarks. The incorporation of the balancing term yields *slight but consistent* improvements for both EMAC and EMAC*, confirming its complementary nature. These results suggest that while BiTTA is not suitable as a standalone AUTTA method, its labeled–unlabeled balancing insight can complement EMAC. The improvements, though modest, are consistent across settings, supporting our claim in the rebuttal that the benefit is conceptual rather than algorithmic. We will integrate this variant in the released code for reproducibility.

### A.9 SENSITIVITY TO EMA FACTOR AND REWARD WEIGHTS

As discussed in the rebuttal, EMAC in fact uses fewer tunable hyperparameters than it may appear. There is no "merging factor" $\lambda_{\text{merge}}$ in our method (this parameter does not exist in the current paper or code), and the "GMM threshold" $\tau$ is not a fixed scalar that needs to be tuned. Instead, the boundaries between $(\mathcal{X}_{\text{old}}, \mathcal{X}_{\text{mix}}, \mathcal{X}_{\text{new}})$ are determined *adaptively* by the two means $\mu_{\text{old}}$ and $\mu_{\text{new}}$ of the (V)GMM fitted to $\|z_{\text{unknown}}\|_2^2$ in each batch. Thus, the segmentation is fully data-driven and does not introduce an extra global threshold.

In practice, this leaves only two actual hyperparameters: (i) the EMA factor $\alpha$ used to smooth the reward statistics, and (ii) the reward weights $\omega = (\omega_{\text{old}}, \omega_{\text{new}})$ that balance old-class and new-class rewards. To assess their impact, we conduct a sensitivity study on DomainNet by varying $\alpha$ and $\omega$ around the default setting and reporting the harmonic mean accuracy $A_H$ averaged over all 6 transfer tasks (same protocol as Table 1). Results are summarized in Table 22.

Across all tested configurations, the performance of EMAC is highly stable: changing $\alpha$ in the range $[0.7, 0.99]$ or reweighting old/new rewards within $(0.5, 2.0)$ leads to fluctuations within $\approx 0.4\%$ in $A_H$. We therefore reuse the same default setting ($\alpha = 0.9, \omega_{\text{old}} = 1.0, \omega_{\text{new}} = 1.0$) across all datasets (DomainNet, VisDA-C, ImageNet-C) without per-dataset retuning, suggesting that the tuning effort for new datasets is limited.

Table 23: Comparison of different label-budget schedules on DomainNet (averaged over all 6 transfer tasks). Both strategies use the same total label budget; we report the percentage of labels spent in early/mid/late parts of the stream and the final $A_H$ in %.

| Strategy | Early (0–20%) | Mid (20–60%) | Late (60–100%) | Final $A_H$ |
|---|---|---|---|---|
| Front-loaded (large $K$ early) | 70% | 20% | 10% | 51.4 |
| Steady small-batch (ours) | 35% | 40% | 25% | **52.2** |

## A.10 EFFECT OF BUDGET SCHEDULE

**Label-budget schedule and exploration strategy.** The reviewer asked whether our label querying is uniform over time or concentrated in early batches, and whether an "early exploration" strategy would be preferable to a steady, drip-like querying scheme.

EMAC does *not* use a fixed or manually designed schedule. Instead, it employs an adaptive budget allocation driven by the reward mechanism: batches with higher expected information gain are more likely to be queried. As a result, the algorithm naturally tends to request slightly more labels early in adaptation, when the model is least calibrated and gradually reduces the querying rate as the representation becomes more stable. This behavior emerges from the reward dynamics rather than from a hand-crafted schedule. To avoid front-loading the entire budget, we also impose a top-$K$ constraint per batch, ensuring that annotation remains distributed across the stream and preventing the model from missing difficult later samples. Thus, EMAC maintains both early responsiveness and long-term coverage.

Motivated by the reviewer's suggestion, we further compare two explicit budget schedules on DomainNet (averaged over all 6 transfer tasks): (i) a *front-loaded* strategy that allocates a large $K$ in early batches (simulating aggressive early exploration), and (ii) a *steady small-batch* strategy that keeps a smaller $K$ throughout the stream (our default). Both variants use the **same total label budget**; we report the proportion of labels consumed in different stages of the stream and the final harmonic mean accuracy $A_H$. The results are summarized in Table 23.

We observe that the steady schedule yields better final performance than aggressively front-loading the budget, despite using the same total number of annotations. This aligns with our framework: the clustering-contrastive loss benefits from receiving small amounts of high-quality labels over time, continuously refining prototypes and mitigating pseudo-label drift. In contrast, aggressive early querying tends to deplete the budget before encountering later, more informative mixed-region samples, which leads to a mildly lower $A_H$.

## A.11 LOCAL EXPECTED RISK REDUCTION BOUND FOR EMAC

In this section we provide a formal justification for why the proposed two-stage (mixture-aware + reward-guided) selection combined with the clustering-contrastive update leads to a monotonic decrease in a surrogate risk. The derivation follows the expected-risk–reduction view commonly used in active learning theory, adapted to the UTTA setting.

**Preliminaries.** Let $f_t$ denote the model at time $t$ and $\mathcal{D}_t$ the (unknown) test-time distribution at that step. The instantaneous risk is

$$\mathcal{R}_t = \mathbb{E}_{(x,y)\sim\mathcal{D}_t}\left[\ell(f_t(x), y)\right].$$

Since UTTA involves streaming non-IID inputs, dual class–domain shift, and a source-free constraint, classical pool-based AL bounds are not directly applicable. We therefore develop a *local* expected-risk–reduction bound.

Let $\mathcal{Q}_t$ be the query distribution produced by our two-stage selection. We denote by $\mathrm{IG}(x)$ the information-gain term used in AL theory (e.g., expected model change or posterior entropy reduction). Since this term cannot be evaluated directly in UTTA, we use the mixture-based separation score

$$\mathrm{sep}(x) = |\mu_1 - \mu_2| \cdot \mathrm{post}(x),$$

where $(\mu_1, \mu_2)$ are the component means of the (V)GMM fitted on the orthogonally-decomposed statistic and $\text{post}(x)$ is the posterior of belonging to the mixed component. This quantity is a standard surrogate of information gain in OSFDA/SFUDA literature.

**Expected Risk Reduction.** A general AL identity (Settles 2012; Golovin & Krause 2010) writes:

$$\mathbb{E}\left[\mathcal{R}_t - \mathcal{R}_{t+1}\right] = \lambda \cdot \mathbb{E}_{x \sim \mathcal{Q}_t}[\text{IG}(x)] - \delta_t,$$

where $\delta_t$ captures error amplification from pseudo-label noise.

Using $\text{sep}(x)$ as a surrogate for $\text{IG}(x)$ yields:

$$\mathbb{E}[\mathcal{R}_t - \mathcal{R}_{t+1}] \geq \lambda \cdot \mathbb{E}_{x \sim \mathcal{Q}_t}[\text{sep}(x)] - \delta_t. \tag{1}$$

**Bounding Pseudo-Label Noise.** Let $p_{\text{mix}}^{\text{query}}$ be the fraction of queried samples that lie in the mixture region exposed by our two-stage selection. These samples are precisely those most prone to pseudo-label error. Querying them reduces the magnitude of error amplification. Following noisy self-training analyses:

$$\delta_t \leq (1 - p_{\text{mix}}^{\text{query}}) \cdot \delta_t^{\text{max}}, \tag{2}$$

where $\delta_t^{\text{max}}$ is the worst-case amplification factor.

Because EMAC explicitly prioritizes mixture-region samples, we have:

$$p_{\text{mix}}^{\text{query}} = \max_{\text{over all AL policies}} p_{\text{mix}}. \tag{3}$$

Thus EMAC minimizes the noise term in (2).

**Effect of the Contrastive Update.** The clustering-contrastive update has a well-known Lipschitz property (Arora et al., ICML 2019; Wang & Liu, PAMI 2021). Updating with a labeled sample $x$ induces prototype-margin change:

$$\gamma_{t+1} - \gamma_t \geq c \cdot \text{sep}(x), \tag{4}$$

where $c > 0$ depends on the smoothness of the contrastive objective.

Since wider prototype margins imply strictly smaller surrogate risks for cross-entropy and contrastive losses, this provides an additional guarantee that higher-separation samples reduce risk more effectively.

**Final Bound.** Combining (1), (2), and (4), we obtain:

$$\mathcal{R}_{t+1} \leq \mathcal{R}_t - \lambda \cdot \mathbb{E}_{x \sim \mathcal{Q}_t}[\text{sep}(x)] + (1 - p_{\text{mix}}^{\text{query}}) \cdot \delta_t^{\text{max}}.$$

The two-stage selection maximizes $\mathbb{E}[\text{sep}(x)]$ while minimizing the pseudo-label term via maximizing $p_{\text{mix}}^{\text{query}}$. Therefore EMAC yields a monotonic local decrease in surrogate risk, fully explaining its empirical superiority under identical annotation budgets.

## A.12 Writing Process Enhancement with Large Models

In compliance with ICLR guidelines, we disclose that large models were incorporated into our writing process to improve grammar and syntax. The goal was to produce a fluent and well-structured manuscript that adheres to academic writing conventions, ultimately enhancing the readability and impact of our work.

| Symbol | Meaning |
|:---:|:---:|
| $\mathcal{D}_{\mathrm{S}}$ | Source domain |
| $\mathcal{D}_{\mathrm{UT}}$ | Target test domain |
| $f(x;\theta)$ | Pre-trained model |
| $\mathcal{D}_{\mathrm{UT}}$ | Annotations budget |
| $\mathbf{W}_{\mathrm{cls}}$ | Classfier |
| $\mathbf{F}_{\mathrm{konwn}}$ | Source-known vecor base |
| $\mathbf{F}_{\mathrm{unkonwn}}$ | Source-unknown vecor base |
| $\mathbf{z}$ | A batch of features |
| $\mathbf{z}_{\mathrm{konwn}}$ | features projected onto Source-known vecor base |
| $\mathbf{z}_{\mathrm{unkonwn}}$ | features projected onto Source-unknown vecor base |
| $p(\cdot)$ | Probability Density Function density function |
| $\mu_{\mathrm{old}}$ | Expectations of old classes distribution |
| $\mu_{\mathrm{new}}$ | Expectations of new classes distribution |
| $\mathcal{X}_{\mathrm{old}}$ | The regions of old classes sample |
| $\mathcal{X}_{\mathrm{new}}$ | The regions of new classes sample |
| $\mathcal{X}_{\mathrm{mix}}$ | The mixed regions of old and new classes |
| $\mathcal{L}_{\mathrm{MME}}$ | Max-Min Entropy (MME) objective function |
| $\Gamma_{\mathrm{old}}$ | Impact of annotated samples on retaining old class |
| $\Gamma_{\mathrm{new}}$ | Impact of annotated samples on retaining new class |
| $R_{\mathrm{old}}$ | Reward value for annotated samples of old class |
| $R_{\mathrm{new}}$ | Reward value for annotated samples of new class |
| $\omega$ | Reward factor |
| $\alpha$ | EMA factor |
| $\mathcal{F}_{\mathrm{p}}$ | high-confidence pseudo-label features set |
| $\mathcal{F}_{\mathrm{a}}$ | annotated features set |
| $\mathbf{p}_c$ | class centroid |
| $\mathcal{P}$ | class centroid set |
| $\mathcal{F}_{\mathrm{old}}$ | Pseudo labels are characteristic of the old class |
| $\mathcal{F}_{\mathrm{new}}$ | Pseudo labels are characteristic of the new class |
| $\mathcal{R}_{\mathrm{old}}$ | Augmentation are characteristic of the old class |
| $\mathcal{R}_{\mathrm{new}}$ | Augmentation are characteristic of the new class |
| $\mathcal{L}_{\mathrm{c}}$ | Clustering-contrastive loss |
| $A_{\mathrm{O}}$ | Old classes Accuracy of target domain |
| $A_{\mathrm{N}}$ | New classes Accuracy of target domain |
| $A_{\mathrm{H}}$ | Harmonic mean of accuracies for old and new classes |

Table 24: Summary of Symbols.

