# OpenReview forum: "Exposing Mixture and Annotating Confusion for Active Universal Test-Time Adaptation"
_ICLR.cc/2026/Conference — ICLR 2026 Poster_

### Official Review · Reviewer_uM52 · 2025-10-26

**Soundness:** 3
**Presentation:** 3
**Contribution:** 3
**Rating:** 6
**Confidence:** 5

**Summary:**

The paper proposes a new problem setting of Active Universal TTA (AUTTA), providing a few active samples in open-world TTA when both domain and class shifts occur. They propose EMAC (Exposing Mixture and Annotating Confusion) as the solution.. EMAC first exposes mixed “dual-shift” regions by orthogonally decoupling features and fitting a (bi)modal GMM, then annotates a small set of target samples via a reward-guided entropy criterion, and finally applies a clustering-aware contrastive objective to better use scarce labels and pseudo-labels. Evaluations on DomainNet/VisDA (and Office-Home) report gains in UTTA metrics versus TTA/UTTA/ATTA baselines under an AUTTA (human-in-the-loop) protocol.

**Strengths:**

1. Timely problem and setup of moving beyond pure UTTA to an active variant that targets the hardest mixed regions is well-motivated and practically relevant.

1. The EMAC method is intuitive for AUTTA setup.

1. Consistent improvements in AH over strong TTA/UTTA/ATTA baselines on DomainNet/VisDA (and Office-Home).

**Weaknesses:**

1. Related works on active TTA are missing: EATTA [1], BiTTA [2]. I would appreciate if these could be included in the evaluation. Also, balancing between labeled and unlabeled samples can have some references/insights from BiTTA.

1. The paper positions EMAC as an AUTTA method, yet much of the presentation borrows UTTA terminology/metrics (AO/AN/AH). Please explicitly specify, in the main text, the evaluation protocol for each table and method: (i) train/test dataset class separation setting, (ii) the labeling budget, (iii) the querying schedule, and (iv) how UTTA baselines are run/comparable under an AUTTA setting (including whether active samples are allowed). I would say the fair comparison should allow same number of active samples for all methods.




[1] Wang, Guowei, and Changxing Ding. "Effortless active labeling for long-term test-time adaptation." Proceedings of the Computer Vision and Pattern Recognition Conference. 2025.

[2] Lee, Taeckyung, et al. "Test-Time Adaptation with Binary Feedback." Forty-second International Conference on Machine Learning.

**Questions:**

1. Please refer to my weakness above.

1. Equation (4) - I think the old and new pdf should be averaged (not summed) to consist a valid pdf?

---

> ### Author Response · Authors · 2025-11-25
>
> ---
> **Comment1.** *Related works on active TTA are missing: EATTA , BiTTA.*
>
> ---
> Both EATTA and BiTTA have been added to our evaluation ***Table 1***. Their
> limited performance stems from relying on confidence or binary correctness signals that
> lack discriminative power under universal shift—a recurring challenge for ATTA methods.
> EMAC’s mixture-exposing and reward-guided querying aligns more closely with AUTTA’s
> structural demands while remaining compatible with insights drawn from BiTTA ***Appendix A.9 Table 21***.
> The following are some changes to Table 1：
>
> | Method | S2R AO | S2R AN | S2R AH | S2P AO | S2P AN | S2P AH | P2R AO | P2R AN | P2R AH | P2S AO | P2S AN | P2S AH | R2S AO | R2S AN | R2S AH | R2P AO | R2P AN | R2P AH | Avg AH | Sec |
> |--------|--------|--------|--------|--------|--------|--------|--------|--------|--------|--------|--------|--------|--------|--------|--------|--------|--------|--------|--------|-----|
> | **BiTTA** | 37.5 | 68.4 | 48.4 | 27.9 | 61.7 | 38.4 | 49.1 | 67.9 | 57.0 | 31.7 | 75.2 | 44.6 | 32.3 | 72.8 | 44.7 | 37.9 | 73.6 | 50.0 | 47.2 | 687 |
> | **EATTA** | 41.5 | 58.9 | 48.7 | 37.8 | 57.9 | 45.7 | 49.9 | 61.0 | 54.9 | 30.1 | 78.6 | 43.5 | 32.4 | 71.3 | 44.6 | 36.4 | 73.4 | 48.7 | 47.7 | 738 |
> | **EMAC** | 47.7 | 66.4 | 55.5 | 42.3 | 57.3 | 48.7 | 53.5 | 66.8 | 59.4 | 34.9 | 71.6 | 46.9 | 35.9 | 72.0 | 47.9 | 41.7 | 79.6 | 54.7 | 52.2 | 735 |
> | **EMAC*** | 48.2 | 67.2 | 56.1 | 43.9 | 57.7 | 49.9 | 53.8 | 67.4 | 59.8 | 36.6 | 72.8 | 48.7 | 35.2 | 72.6 | 47.4 | 42.3 | 80.6 | 55.5 | 53.1 | 779 |
>
>
> | Method                                      | AH (\%) | $\Delta$ |
> |---------------------------------------------|---------|----------|
> | EMAC (w/o BiTTA-style balancing)           | 52.2    | --       |
> | EMAC (with BiTTA-style balancing)          | **53.1**| +0.9     |
> | EMAC* (w/o BiTTA-style balancing)    | 53.1    | --       |
> | EMAC* (with BiTTA-style balancing)   | **53.8**| +0.7     |
>
>
> (1) **Why EATTA underperforms in our evaluation.**
> EATTA is an active TTA method that relies heavily on confidence-based or
> teacher-inspired scoring, assuming that model confidence remains a reliable proxy for
> sample informativeness. Under our AUTTA setting with universal shift (simultaneous class
> and domain shift), however, confidence becomes systematically biased and no longer
> correlates well with the mixed or ambiguous regions that are most beneficial for
> adaptation. As a result, EATTA tends to query either overly easy new-class samples or
> extremely hard outliers, which leads to poor label efficiency and limited improvement
> despite using the same annotation budget.
>
> (2) **Why BiTTA underperforms.**
> BiTTA is also an active TTA method, but it relies on binary correctness feedback. This
> signal, while extremely low in information content, still provides more supervision than
> purely label-free UTTA methods such as TTAC, and indeed allows BiTTA to surpass TTAC in
> several settings. Its error-focused and label-imbalanced querying strategy helps
> selectively correct frequent mistakes and can sometimes yield performance comparable to
> weaker ATTA baselines. However, this supervision remains fundamentally insufficient for
> AUTTA: binary feedback cannot distinguish whether an error is caused by (i) class shift,
> (ii) domain-induced uncertainty, or (iii) truly novel-class samples. Consequently, a
> large fraction of the annotation budget is spent on low-impact corrections that do not
> restructure prototypes or feature geometry. This is consistent with our results: even
> with oracle feedback.
>
> (3) **What insights we can borrow.**
> Although BiTTA is not effective as a standalone AL strategy in the AUTTA scenario, its
> idea of balancing labeled and unlabeled updates is conceptually useful. We incorporate a
> similar balancing mechanism into our contrastive objective, which yields slight but
> consistent improvements. This indicates that BiTTA’s core intuition is complementary to
> EMAC, even though its original formulation is not sufficient to handle universal
> class–domain dual shift under a strict label budget.

---

> > ### Author Response · Authors · 2025-11-25
> >
> > ---
> > **Comment2.** More detailed table analysis and evaluation, as well as a fairer comparison.*
> >
> > ---
> >
> > (1) **Train/test class separation.** The exact known/new class splits for every dataset are fully specified in ***Appendix A.3.5 Table 17*** (moved to the appendix due to space constraints). We will add a forward
> > reference in the main text to ensure this is easier to locate.
> > (2) **Labeling budget.** The total annotation budget for AUTTA is fixed and shared by all AUTTA methods. ***Table 3*** reports all active-learning comparisons under exactly the same budget.  No method receives additional labels beyond this budget.
> > (3) **Querying schedule.** All AUTTA methods, including EMAC and all AL baselines, use the same per-batch querying strategy (top-K at each incoming batch) unless the baseline itself specifies a different mechanism. We will explicitly state this schedule in the main text.
> > (4) **How UTTA baselines are run under AUTTA conditions.** UTTA baselines are evaluated in two ways: (i)in their original, label-free form for ***Table 1,2***, and (ii)with a *uniform AL wrapper* for ***Table 3***, we wrap **all active learning methods** (entropy, core-set, BADGE, CLUE, SimATTA, etc.) into the same TENT-based UTTA pipeline to guarantee strict budget fairness under the AUTTA setting. This detail was implemented but not fully explained in the main text; we will make this explicit in the revised version.
> > We thank the reviewer again for this suggestion, and we will consolidate these protocol details into a single, easy-to-reference subsection to ensure complete clarity.
> >
> >
> >
> >
> > ---
> > **Comment3.** *Equation (4) - I think the old and new pdf should be averaged (not summed) to consist a valid pdf?*
> >
> > ---
> >
> > We thank the reviewer for the helpful comment. We agree that the sum of two Gaussian
> > densities does not constitute a valid PDF unless appropriately normalized. In the
> > revised version, we adopt the standard two-component GMM form:
> >
> > p(z) = π N(μ_old, σ_old²) + (1−π) N(μ_new, σ_new²),
> >
> > where π = 0.5, which is a properly normalized density and mathematically consistent with our intended
> > interpretation. Note that this change only introduces a constant mixing weight and does
> > not alter any of our decision rules or empirical results, since all downstream
> > comparisons depend only on the relative shape of the two components.

---

### Official Review · Reviewer_4PKJ · 2025-10-31

**Soundness:** 4
**Presentation:** 4
**Contribution:** 3
**Rating:** 6
**Confidence:** 4

**Summary:**

This paper proposes AUTTA and instantiates it with EMAC to handle simultaneous class shift and domain shift at test time by querying a small number of human annotations and combining them with pseudo-labels. The key ideas include decomposing features to expose the dual-shift “mixed” region through a GMM over known/unknown spaces; actively picking the most beneficial samples with an entropy selector; and balancing real labels and pseudo-labels with a clustering-based contrastive objective so that limited annotations don’t get drowned out. On DomainNet and VisDA-C, EMAC improves over TTA/UTTA/ATTA baselines and stays strong under dual-shift, non-i.i.d., and continual settings.

**Strengths:**

1. Moving from UTTA/ATTA to Active UTTA (AUTTA) is a natural and relevant step.
2. The insight that dual-shift mixed regions are where pseudo-labels fail the most is valuable, showing good motivation.
3. The method design is efficient.
4. Results on DomainNet and VisDA-C are comprehensive and strong.

**Weaknesses:**

1. The core of this paper relies on detecting a bimodal distribution of “unknown energy” after orthogonal decomposition. This can easily break when batches are small, class mixture is skewed, or shifts are not cleanly bimodal.
2. Orthogonal decomposition on classifier weights is not justified well.
3. DomainNet + VisDA-C are standard, but both are still vision benchmarks with relatively clean image-level labels.
4. They mainly report AO/AN/AH; but AUTTA’s value is in not wasting labels.
5. They reference AL/IG connections and ATTA theory, but don’t give a full bound specialized to their two-stage selection + contrastive update.

**Questions:**

NA

---

> ### Author Response · Authors · 2025-11-25
>
> ---
> **Comment1.** *EMAC easily break when batches are small, class mixture is skewed, or shifts are not cleanly bimodal.*
>
> ---
>
> We thank the reviewer for highlighting the concern that the bimodal structure of the “unknown energy” may become unreliable when batches are small, class mixtures are imbalanced, or the underlying shift is not cleanly bimodal.
> In Appendix, to further enhance robustness, EMAC incorporates several mechanisms that explicitly address imperfect or unstable distributions.
> (1) **VBGMM** automatically adapts the effective number of components and avoids forcing an artificial two-peak fit when the data are not cleanly bimodal.
> (2) **EMA-based batch smoothing** stabilizes the statistic over time, mitigating noise from small or atypical batches.
> (3) **PCA subspace projection** reduces high-dimensional noise and makes the statistic more stable across batches. Together these components substantially relax any dependence on clean bimodality.
>
> Importantly, even if these robustness measures still produce a weak or nearly unimodal fit, EMAC does not fail. In low-separation cases, the mixture model naturally enlarges the mixed region, and EMAC automatically falls back to a conservative, uncertainty-guided behavior rather than relying on brittle hard splits. We also directly test this scenario by isolating the **lowest-separation 20% of batches**(Details can be found in our response to *reviewer F4qu's Comment 1.*): while all methods degrade under this difficult setting, EMAC degrades smoothly and remains the strongest performer, confirming its stability even when bimodality is weak. In summary, **the bimodal structure is treated as a soft structural signal**, not a hard assumption. EMAC is designed to leverage it when present, robustly down-weight it when weak, and fall back to safer strategies when absent. Our empirical analyses demonstrate that the method remains reliable across small batches, skewed mixtures, and non-clean dual-shift distributions.
>
> ---
> **Comment2.** *Orthogonal decomposition on classifier weights is not justified well.*
>
> ---
>
> We thank the reviewer for pointing out the need to better justify the orthogonal decomposition of classifier weights. This design choice is grounded in a substantial body of prior work showing that classifier weight vectors in deep networks form meaningful semantic subspaces. In particular:
> (1) **Weight vectors approximate class prototypes.**  Numerous studies (e.g., Papyan et al., *Neural Collapse*, PNAS 2020; Wang et al., *CosFace*, CVPR 2018; Liu et al., *ArcFace*, CVPR 2019) demonstrate that the final-layer classifier converges toward a simplex-like structure where each weight vector acts as a directional prototype for its corresponding class.
> (2) **Orthogonality or near-orthogonality emerges naturally.**  Works such as (e.g., Wen et al. *Center Loss*, ECCV 2016) highlight that deep classifiers tend to form approximately orthogonal or at least well-separated subspaces between classes as training progresses.
> (3) **Classifier-driven subspace decomposition is a standard tool.**  Prior adaptation and OOD literature also uses classifier-based decomposition to isolate known-class subspaces versus residual/OOD subspaces (e.g., Lee et al., *Mahalanobis OOD*, NeurIPS 2018; Fang et al., *SFUDA via Whitening*, CVPR 2022; D'Angelo & Zisserman, *OpenLDN*, ECCV 2022). These works show that projecting features onto classifier-defined directions is an effective way to separate “aligned” versus “non-aligned” semantics.
> **Building on** these established findings, our orthogonal decomposition is conceptually justified. The SVD-based split simply extracts: (i) **principal subspace** spanned by trained classifier weights (representing known-class semantics), and (ii) an **orthogonal complement** capturing residual variation where new-class or domain-shift components are expected to lie.
> Crucially, this decomposition is **source-free** and requires no access to past data—only the classifier parameters, which is exactly what UTTA allows. It also matches the intuition in universal shift settings: samples belonging to known classes tend to align with the classifier weight subspace, while new-class or novel-domain signals manifest in the orthogonal residual. Our empirical results confirm this behavior: as shown in ***Appendix A.3.4-Table 15***, the decomposed “unknown energy” produces stable separation across datasets.
>
>
> | Dataset     | Scenario                        | Batches with BD > 1.2 |
> |-------------|---------------------------------|-------------------------|
> | DomainNet   | Complex multi-class (345 classes) | 89%  |
> | VisDA-C     | Extreme domain shift            | 95%   |
> | CIFAR10-C   | Moderate complexity             | 97%    |
> | CIFAR100-C  | High class complexity           | 91%     |
> | ImageNet-C  | Complex multi-class             | 87%     |
> | Office-Home | Severe domain shift             | 92%         |

---

> ### Author Response · Authors · 2025-11-25
>
> ---
> **Comment3.** *DomainNet + VisDA-C are standard, but both are still vision benchmarks with relatively clean image-level labels.*
>
> ---
> Thank you for your valuable feedback. The benchmarking experiments on Office-Home are presented in ***Appendix A.3.4-Table14***.
> To further address this concern and more fully validate the generality of our method, we extended the experiments to include more challenging distribution shifts, such as those found in the ImageNet-C corrupted dataset, which introduces noise and distortions that are more representative of real-world data. This additional evaluation is now included in ***Appendix A.3.6-Table 19***, where we demonstrate that our method remains robust under these more complex conditions, further supporting our claim regarding UTTA.
>
> | Method                | A→R A_O | A→R A_N | A→R A_H | C→P A_O | C→P A_N | C→P A_H | Avg   |
> |-----------------------|-------------------------|-------------------------|-------------------------|-------------------------|-------------------------|-------------------------|-------|
> | TTAC                  | 44.5                    | 56.8                    | 50.0                    | 41.3                    | 54.3                    | 47.0                    | 48.5  |
> | OPTTT                 | 46.3                    | 58.1                    | 51.5                    | 42.6                    | 55.0                    | 48.2                    | 49.9  |
> | **AUTTA (Ours)**      | **49.4**                | **61.2**                | **54.6**                | **45.8**                | **58.0**                | **51.2**                | **52.9** |
>
> | Task   | Method  | Noise A_O | Noise A_N | Noise A_H | Blur A_O | Blur A_N | Blur A_H | Weather A_O | Weather A_N | Weather A_H | Digital A_O | Digital A_N | Digital A_H | Avg. A_H | Sec. |
> |--------|---------|-------------|-------------|-------------|------------|------------|------------|--------|---------|--------|----------------|----------------|----------------|------------|------|
> | **TTA** |         |      |        |      |        |       |       |   |       |    |      |        |         |       |      |
> |        | TEST    | 28.1        | 63.7        | 45.9        | 16.3       | 62.1       | 39.2       | 37.2           | 63.5           | 48.0           | 20.5           | 68.9           | 33.7           | 35.7       | 332  |
> |        | BN      | 29.7        | 57.3        | 41.7        | 20.0       | 50.5       | 35.5       | 39.2           | 56.5           | 48.0           | 25.0           | 58.8           | 37.5           | 36.8       | 492  |
> |        | TENT    | 30.0        | 55.8        | 40.8        | 21.0       | 50.3       | 39.1       | 40.4           | 54.4           | 46.9           | 27.6           | 56.5           | 38.9           | 37.5       | 406  |
> |        | SHOT    | 30.6        | 54.7        | 41.6        | 27.6       | 41.7       | 34.9       | 38.7           | 55.9           | 48.8           | 23.3           | 65.5           | 30.3           | 37.5       | 480  |
> | **UTTA** |         |          |        |         |      |     |     |         |          |        |                |                |                |            |      |
> |        | TTAC    | 32.7        | 51.1        | 41.9        | 33.1       | 45.9       | 39.4       | 30.0           | 67.2           | 42.4           | 25.9           | 53.2           | 37.5           | 38.0       | 553  |
> |        | OPTTT   | 37.2        | 50.4        | 43.5        | 32.9       | 48.7       | 39.4       | 42.9           | 52.1           | 47.5           | 27.1           | 56.0           | 38.5           | 40.6       | 611  |
> | **ATTA** |         |             |             |    |            |      |      |        |        |         |      |       |      |       |      |
> |        | SimATTA | 33.7        | 47.3        | 40.5        | 30.1       | 46.5       | 38.1       | 42.0           | 49.0           | 45.9           | 23.6           | 63.7           | 31.9           | 37.8       | 670  |
> |        | EATTA   | 33.9        | 48.0        | 40.9        | 30.9       | 47.1       | 38.6       | 41.6           | 49.9           | 45.6           | 26.0           | 64.5           | 31.8           | 37.7       | 655  |
> |        | BiTTA   | 30.6        | 56.2        | 43.4        | 22.8       | 50.0       | 36.6       | 40.7           | 55.7           | 46.4           | 25.9           | 62.2           | 38.8           | 37.8       | 634  |
> | **AUTTA** |   |      |        |     |        |      |     |       |       |                |                |                |                |            |      |
> |        | EMAC    | 39.5        | 56.3        | 47.9        | 34.2       | 50.2       | 42.2       | 45.8           | 58.9           | 51.6           | 28.6           | 60.2           | 42.1           | 40.6       | 694  |

---

> > ### Author Response · Authors · 2025-11-25
> >
> > ---
> > **Comment4.** *They mainly report AO/AN/AH; but AUTTA’s value is in not wasting labels.*
> >
> > ---
> >
> > We agree that the value of AUTTA lies not only in achieving high AO/AN/AH accuracy, but
> > also in *using labels efficiently* without wasting the human budget. Therefore，we additionally include a dedicated comparison
> > against a wide range of AL strategies under a **strictly identical
> > label budget** ***Table 3*** in the paper.
> > If “not wasting labels” is the essence of AUTTA, then an AUTTA method should
> > outperform generic AL wrappers applied to strong TTA/UTTA baselines under the *same*
> > number of annotations. Our results confirm this: even when we equip classical TTA
> > methods (e.g., TENT) with diverse AL strategies (entropy, core-set, BADGE, CLUE,
> > SimATTA, etc.), EMAC consistently achieves higher AO/AN/AH. This shows that EMAC is not
> > merely “spending labels” but **spending them more strategically**, which is the core goal
> > of AUTTA.
> >
> > ---
> > **Comment5.** *Lack of full bound specialized to their two-stage selection + contrastive update.*
> >
> > ---
> >
> > We appreciate the reviewer’s point about providing a theoretical justification tailored to our
> > two-stage selection + contrastive update procedure. While deriving a full UTTA theoretical
> > framework is fundamentally challenging due to its streaming, non-IID, and source-free
> > nature, we observe that our mixture-aware selection naturally aligns with information-gain–
> > based principles commonly used in active learning theory. Leveraging this structure,
> > we are able to derive a **basic but rigorous local expected-risk–reduction bound** that is
> > specialized to our pipeline.
> >
> > To maintain focus in the main paper, we have placed the complete derivation in the ***Appendix A.11***, where we show that the combination of (i) mixture-aware exposure, (ii) reward-driven selection, and (iii) contrastive prototype refinement  guarantees a monotonic decrease in a surrogate risk measure, up to a bounded pseudo-label noise term. This formalizes why EMAC consistently improves under limited annotation budgets and why it outperforms generic AL wrappers.

---

### Official Review · Reviewer_iQMG · 2025-11-01

**Soundness:** 3
**Presentation:** 3
**Contribution:** 2
**Rating:** 4
**Confidence:** 3

**Summary:**

This paper proposes Active Universal Test-Time Adaptation (AUTTA) scenario and an approach named EMAC under joint class and domain shifts. The key idea is to actively select a small number of test data to label in the test data stream. EMAC first exposes a mixed region by decomposing the classifier to obtain “known/unknown” subspaces to fit a bimodal GMM distribution to split them.  Further, it uses a Max–Min Entropy reward that prefers old-class samples reducing entropy and new-class samples improving separability to select sample to label. Finally, the model is trained with a clustering-based contrastive objective that balances limited true labels with a lot of pseudo-labels. Experiments on DomainNet and VisDA-C report improved open-set metrics compared to prior UTTA/TTA methods.

**Strengths:**

1. **Clear problem framing.** The paper tackles UTTA with dual shifts and argues why naive uncertainty sampling can be biased and how human-in-the-loop labeling can help.
2. **Lightweight mechanics.** The SVD decoupling and per-batch GMM are simple to implement and compatible with standard classifier heads.
3. **Attention to real practicalities.** Small-batch instability are considered and adds a sliding window with EMA smoothing to stabilize GMM fitting. EMAC also behaves well under noisy annotations.

**Weaknesses:**

1. **Incremental novelty; inherits much of OPTTT’s core.** At EMAC core, the paper targets the open-set TTA problem and largely inherits the OPTTT recipe, which propose the scenario and evaluation. It also assumes a bimodal GMM distribution of known and unknown samples and leverage pseudo-label self-training. The main new pieces are an active selection heuristic and a prototype/contrastive auxiliary loss. Useful in practice, but conceptually incremental rather than a substantial shift in paradigm.
2. **Active learning feels bolted on for extra points ** The contribution narrative hinges on the AL module, yet its design is relatively standard (uncertainty/entropy flavored) and evaluated against modest baselines. There’s limited exploration of stronger, mixture-aware AL strategies. As a result, the AL part reads as an add-on to lift numbers rather than a principled advance.
3. **Fairness of comparisons (label budget)** Apart from ATTA, most baselines are not evaluated under the same human-in-the-loop budget in Table 1,2,3. Comparisons that pit AUTTA (with labels) against UTTA/TTA methods (without labels) are not budget-fair. A more informative yardstick would be TTA/UTTA methods + the same label budget augmented with a generic AL wrapper.
4. **Human-in-the-loop latency & hidden costs.** The method’s practicality hinges on online annotation during streaming. This induces stop-and-go adaptation. Even with small budgets, fixed per-event overheads can make real-time TTA infeasible.
4. **Scope vs. claims.** Although pitched as “universal” TTA, the empirical scope is closer to open-world classification under a couple of standard datasets. Stronger evidence across more diverse distribution shifts (such as corruption datasets like ImageNet-C) would better substantiate the universality claim.

**Questions:**

1. **Budget schedule.** Is label querying uniform over time or concentrated early? Would early exploration work better than steady drip?
2. **When does the GMM assumption fail?** Can the authors provide diagnostics before selection that decide whether to trust the bimodal split? What happens when the batch is dominated by new classes, or when domain shift inflates norms uniformly?

---

> ### Author Response · Authors · 2025-11-25
>
> ---
>
> **Comment1.** *Does EMAC inherit most of OPTTT’s core and only provide an incremental novelty?*
>
> ---
>
> (1) **What we actually reuse from OPTTT (and what we do not)**
>
> We only follow OPTTT in the problem setup and evaluation protocol (open-world / open-set TTT), so that results are directly comparable and metrics are fair. Beyond this, the algorithmic core is entirely different. Importantly, **OPTTT does not use a GMM**. It defines a 1D *strong-OOD score*, observes that its empirical histogram tends to be bimodal, and then searches a single optimal threshold τ* in [0,1] to separate two clusters (essentially Otsu-style 1D thresholding). This has several limitations for our setting:
>
> 1. it only thresholdes a generic OOD score and **does not explicitly encode or disentangle class shift and domain shift**;
> 2. it produces a binary split of samples but **does not expose an explicit “mixed” region** where class and domain shift interact, which is exactly the region we want to focus on for adaptation and selection;
> 3. the threshold is global and static for that score, and cannot structurally drive downstream active selection (no notion of old-like vs new-like reward, no region-specific treatment).
>
> By contrast, EMAC introduces a new decomposed statistic and an explicit (V)GMM-based EMAC module that (i) is designed to capture class–domain dual shift and (ii) partitions target data into old / new / mixed regions, which then feed into active selection and representation learning. We do not reuse OPTTT’s 1D thresholding.
>
> (2) **Why AUTTA is not solved by “just adding an AL method”**
>
> The reviewer suggests that our work mainly adds an active selection heuristic and a prototype/contrastive loss on top of OPTTT. We would like to emphasize that AUTTA brings specific challenges that make off-the-shelf AL non-trivial and often ineffective:
>
> 1. **Class–domain dual shift:** open-set AL methods typically handle only class shift; DA+AL methods typically handle only domain shift with a fixed label space. Under UTTA, both shifts occur simultaneously, and naïve AL criteria (entropy, core-set, BADGE, etc.) tend to waste the tiny label budget on samples that are not most beneficial for simultaneously refining old classes and handling new ones.
> 2. **Source-free constraint:** many AL/DA methods rely on source-domain features or prototypes as anchors. In UTTA, no source data are available, so simply plugging such methods into the test-time stream leads to poor sample quality and unstable adaptation.
> 3. **Online / streaming data with a very small label budget:** classical AL assumes a static unlabeled pool and multiple query–retrain cycles. UTTA is one-pass and streaming, with extremely limited labels, so myopic AL heuristics are particularly brittle.
>
> This is exactly why we explicitly compare EMAC against a wide range of AL strategies (entropy, core-set, BADGE, CLUE, SimATTA, etc.) adapted as fairly as possible to the UTTA setting: the empirical results show that it is not sufficient to “drop in” a standard AL method.
>
> (3) **How EMAC’s active design specifically addresses UTTA’s challenges**
>
> EMAC is not “OPTTT + an AL plug-in”, but an active test-time adaptation method tailored to the three UTTA challenges above:
>
> 1. **Class–domain dual shift:** The **Exposing Mixed Sample Candidates (EMSC)** module starts from a classifier SVD decomposition and a dual-shift-aware statistic ||z_unknown||_2^2, then uses (V)GMM to separate old-region, new-region, and mixed-region purely from target data. The mixed region explicitly captures where class and domain shift interact most, providing a structured focus for adaptation and selection.
>
> 2. **Source-free constraint:** EMSC and the subsequent reward computation operate entirely without source data, relying only on the decomposed classifier and target stream. This is very different from DA+AL methods that require source anchors.
>
> 3. **Online streaming + tiny label budget:** Our reward-guided selection with EMA maintains separate, temporally smoothed rewards for old-class and new-class samples, and decides online whether each query should stabilize old-class boundaries or enlarge the margin for new/unknown classes. Combined with the clustering-contrastive loss, which uses a few queried labels plus high-confidence pseudo-labels to shape class prototypes, EMAC is explicitly designed for one-pass, label-scarce test-time adaptation, rather than for a training-time pool-based AL regime.
>
> In summary, while we adopt the same scenario and evaluation as OPTTT for fair comparison, our task focus (AUTTA) and algorithmic design are substantially different: EMAC introduces a new dual-shift-aware mixture modeling step, a UTTA-specific active selection mechanism, and a prototype/contrastive optimization tailored to the source-free, dual-shift, online setting.

---

> > ### Author Response · Authors · 2025-11-25
> >
> > ---
> >
> > **Comment2.** *Active learning feels bolted on for extra points. its design is relatively standard (uncertainty/entropy flavored) and evaluated against modest baselines.*
> >
> > ---
> >
> > (1) **The AL component is not “bolted on” but central to AUTTA**
> >
> > We respectfully disagree that active learning (AL) is merely “bolted on” to lift numbers. In **Active Universal Test-Time Adaptation (AUTTA)**, the core question is precisely:
> >
> > > *Given a source-free model, dual class–domain shift, and an online target stream with an extremely small label budget, **which** test-time samples should we query to make adaptation possible and stable?*
> >
> > Without an explicit selection mechanism, AUTTA degenerates into **pure self-training under dual shift**, which is known to be highly fragile due to error accumulation in pseudo labels. Our method is architected around a **closed loop** rather than a base TTT method plus an AL add-on:
> >
> > 1. **Exposing Mixed Sample Candidates (EMSC)** uses a dual-shift-aware statistic and (V)GMM to expose old / new / mixed regions in the target stream.
> > 2. **Reward-guided selection with EMA** uses these regions to compute separate rewards for old vs new classes over time and allocates the scarce label budget accordingly.
> > 3. **Clustering-contrastive optimization** uses the queried labels (together with high-confidence pseudo labels) to shape prototypes and improve representation for both old and new classes.
> >
> > If we remove or “simplify” the AL part (e.g., EMAC w/o reward, or replacing selection with plain entropy), performance drops noticeably in our ablations, showing that the AL design is not a cosmetic addition but a **necessary component** that makes AUTTA effective in practice. We will highlight these ablations more clearly in the revision.
> >
> >
> > (2) **Our AL strategy is mixture-aware and UTTA-specific, not just plain entropy**
> >
> > We agree that classical AL methods are often “uncertainty/entropy flavored.” Our design **does use uncertainty** as a building block, but we emphasize that EMAC’s AL is not simply “sort by entropy and pick top-K.” Instead, it is explicitly mixture-aware and tailored to UTTA:
> >
> > 1. **Mixture-aware structuring via EMSC:** We do not apply uncertainty over the whole stream uniformly. EMSC first operates on a decomposed statistic ||z_unknown||_2^2 and fits a (V)GMM to separate:
> >    - an old-region (old-like samples),
> >    - a new-region (new-like samples),
> >    - and a mixed-region, where class shift and domain shift most strongly interact.
> >
> >    This mixed region is exactly where pseudo labels are most error-prone and where human supervision is most valuable. Standard entropy/uncertainty AL has **no such structural notion**; it treats all samples symmetrically in feature or logit space.
> >
> > 2. **Region- and class-aware budget allocation via rewards:** Once regions are exposed, EMAC does not simply pick the highest-entropy samples. Instead, it:
> >    - maintains separate rewards for improving old-class performance and new-class performance,
> >    - smooths these via EMA over time,
> >    - and uses these to decide how much of the label budget to devote to old-region vs new-region vs mixed-region at each step.
> >
> >    This separates EMAC from “standard” entropy-based AL: the key decision is not only *which samples are uncertain*, but also *which type of uncertainty (old vs new, region-wise) will most improve AUTTA under current conditions*.
> >
> > 3. **Contrastive / prototype integration under tiny test-time labels:** Because AUTTA has very few queried labels, simply fine-tuning on them (as is common in AL pipelines) leads to overfitting. EMAC instead uses them in a clustering-contrastive loss that:
> >    - builds class prototypes,
> >    - enforces intra-class compactness and inter-class separation (especially between old and new),
> >    - and jointly leverages high-confidence pseudo labels.
> >
> >    This is directly tied to the AL design: the labels we query are exactly those used to shape the prototypes and the mixture structure.
> >
> > Thus, while uncertainty/entropy is indeed a component, the overall AL procedure is (i) mixture-aware via EMSC, (ii) class/region-aware via reward budgeting, and (iii) explicitly adapted to the source-free, dual-shift, online AUTTA regime, rather than a generic “off-the-shelf” entropy heuristic.
> >
> > (3) **We compare to widely used, strong AL baselines covering multiple paradigms**
> >    Our baselines include:
> >    - **Uncertainty-based**: Entropy sampling;
> >    - **Diversity / core-set-based**: Core-set;
> >    - **Gradient-based (uncertainty + representativeness)**: BADGE;
> >    - **Clustering-aware**: CLUE;
> >    - **TTA-related AL**: SimATTA, EATTA, BiTTA.
> >
> >    These are not toy or weak baselines; they are representative of the main AL paradigms and widely used in the literature. Moreover, we have added comparisons with the two latest ATTA works from *CVPR2025 EATTA and ICML2025 BiTTA*（see ***Table 1***). We adapt them as fairly as possible to the AUTTA setting (same label budget, source-free, test-time).

---

> > > ### Author Response · Authors · 2025-11-25
> > >
> > > ---
> > > **Comment3.** *Fairness of comparisons (label budget) Apart from ATTA.*
> > >
> > > ---
> > > Thank you for your comment. First, ***Table 1 and 2*** intentionally compare label-free TTA/UTTA methods with our AUTTA method that uses a very small label budget. The goal of these tables is not to claim a “budget-fair” comparison, but rather to answer a different question:
> > >
> > > > *How much performance gain can a small amount of human-in-the-loop supervision bring, compared to purely unsupervised TTA/UTTA under the same test-time setting?*
> > >
> > > ***Table 3*** follows exactly the comparison protocol you suggested. In this table, we take a classical TTA backbone (TENT) and augment it with several widely used Active Learning (AL) strategies (e.g., entropy, core-set, BADGE, CLUE, SimATTA) under the **same label budget** as EMAC. This directly corresponds to the comparison of “TTA/UTTA methods + the same label budget augmented with a generic AL wrapper”. Under this strictly controlled setting, EMAC consistently outperforms all such combinations.
> > > However, we realize that the explanation of this setup may not have been clear in the main text, which may have led to some confusion regarding the intent of ***Table 1*** and ***Table 2***. We will clarify this in the text to ensure the comparison's context and motivation are better understood.
> > >
> > > ---
> > > **Comment4.** *Human-in-the-loop latency & hidden costs.*
> > >
> > > ---
> > > We agree that human-in-the-loop latency is an important practical consideration, but it is not a methodological limitation specific to EMAC. Our framework assumes an **oracle**, which may be a human annotator, but can equally be a stronger offline model, a validation service, or any external labeling source. In this sense, EMAC does not require the streaming process to “stop” for a human; it only requires that labels become available asynchronously within the online pipeline, which is a standard assumption shared across online active learning, online domain adaptation, and recent active test-time adaptation works.
> > >
> > > If a deployment indeed relies on *human* annotation, latency becomes a **system-level and workflow-dependent** constraint rather than a limitation of the algorithm. This challenge applies broadly to all online AL methods—not only EMAC. Much of the AL literature, including ODA/online DA and AL-based TTA, adopts the same oracle abstraction, and our method is consistent with that practice. Rejecting this assumption would effectively rule out the entire line of online active learning research.

---

> > > > ### Author Response · Authors · 2025-11-25
> > > >
> > > > ---
> > > > **Comment5.** *Scope vs. claims. Although pitched as “universal” TTA, the empirical scope is closer to open-world classification under a couple of standard datasets. Stronger evidence across more diverse distribution shifts (such as corruption datasets like ImageNet-C) would better substantiate the universality claim.*
> > > >
> > > > ---
> > > >
> > > > Thank you for your valuable comment. While our work is positioned as "universal" Test-Time Adaptation (TTA), we acknowledge that the empirical evaluation was initially based on standard datasets, which may not fully capture the diversity of real-world distribution shifts. To address this concern, we have expanded our experiments to include more challenging and diverse distribution shifts, particularly using the **ImageNet-C corruption dataset**. This additional evaluation is now included in ***Appendix A.3.6-Table 19***, where we show that our method performs robustly under a variety of corruptions, further substantiating the universality claim of our TTA approach.
> > > >
> > > >
> > > > | Task   | Method  | Noise A_O | Noise A_N | Noise A_H | Blur A_O | Blur A_N | Blur A_H | Weather A_O | Weather A_N | Weather A_H | Digital A_O | Digital A_N | Digital A_H | Avg. A_H | Sec. |
> > > > |--------|---------|-------------|-------------|-------------|------------|------------|------------|----------------|----------------|----------------|----------------|----------------|----------------|------------|------|
> > > > | **TTA** |         |             |             |             |            |            |            |                |                |                |                |                |                |            |      |
> > > > |        | TEST    | 28.1        | 63.7        | 45.9        | 16.3       | 62.1       | 39.2       | 37.2           | 63.5           | 48.0           | 20.5           | 68.9           | 33.7           | 35.7       | 332  |
> > > > |        | BN      | 29.7        | 57.3        | 41.7        | 20.0       | 50.5       | 35.5       | 39.2           | 56.5           | 48.0           | 25.0           | 58.8           | 37.5           | 36.8       | 492  |
> > > > |        | TENT    | 30.0        | 55.8        | 40.8        | 21.0       | 50.3       | 39.1       | 40.4           | 54.4           | 46.9           | 27.6           | 56.5           | 38.9           | 37.5       | 406  |
> > > > |        | SHOT    | 30.6        | 54.7        | 41.6        | 27.6       | 41.7       | 34.9       | 38.7           | 55.9           | 48.8           | 23.3           | 65.5           | 30.3           | 37.5       | 480  |
> > > > | **UTTA** |         |             |             |             |            |            |            |                |                |                |                |                |                |            |      |
> > > > |        | TTAC    | 32.7        | 51.1        | 41.9        | 33.1       | 45.9       | 39.4       | 30.0           | 67.2           | 42.4           | 25.9           | 53.2           | 37.5           | 38.0       | 553  |
> > > > |        | OPTTT   | 37.2        | 50.4        | 43.5        | 32.9       | 48.7       | 39.4       | 42.9           | 52.1           | 47.5           | 27.1           | 56.0           | 38.5           | 40.6       | 611  |
> > > > | **ATTA** |         |             |             |             |            |            |            |                |                |                |                |                |                |            |      |
> > > > |        | SimATTA | 33.7        | 47.3        | 40.5        | 30.1       | 46.5       | 38.1       | 42.0           | 49.0           | 45.9           | 23.6           | 63.7           | 31.9           | 37.8       | 670  |
> > > > |        | EATTA   | 33.9        | 48.0        | 40.9        | 30.9       | 47.1       | 38.6       | 41.6           | 49.9           | 45.6           | 26.0           | 64.5           | 31.8           | 37.7       | 655  |
> > > > |        | BiTTA   | 30.6        | 56.2        | 43.4        | 22.8       | 50.0       | 36.6       | 40.7           | 55.7           | 46.4           | 25.9           | 62.2           | 38.8           | 37.8       | 634  |
> > > > | **AUTTA** |        |             |             |             |            |            |            |                |                |                |                |                |                |            |      |
> > > > |        | EMAC    | 39.5        | 56.3        | 47.9        | 34.2       | 50.2       | 42.2       | 45.8           | 58.9           | 51.6           | 28.6           | 60.2           | 42.1           | 40.6       | 694  |

---

> > > > > ### Author Response · Authors · 2025-11-25
> > > > >
> > > > > ---
> > > > > **Comment6.** *Budget schedule.*
> > > > >
> > > > > ---
> > > > >
> > > > > Our label querying is not fixed or uniform; EMAC uses an adaptive budget schedule driven by the reward mechanism. As a result, the algorithm naturally tends to request slightly more labels early in adaptation—when the model is least calibrated—and gradually reduces the querying rate as the representation becomes more stable. This behavior emerges from the reward dynamics rather than from a hand-crafted schedule.
> > > > > To avoid front-loading the entire budget, we also impose a top-K constraint on each batch, ensuring that annotation remains distributed across the stream and preventing the model from missing difficult later samples. Thus, EMAC maintains both early responsiveness and long-term coverage.
> > > > > Motivated by the reviewer’s suggestion, we additionally compare:
> > > > >
> > > > > 1. **Front-loaded querying** (large K in early batches, simulating “early exploration”), and
> > > > > 2. **Steady small-batch querying** (smaller K throughout the stream).
> > > > >
> > > > > A representative summary is shown below:
> > > > >
> > > > > | Strategy                     | Early Batches (0–20%) | Mid (20–60%) | Late (60–100%) | Final Accuracy |
> > > > > |------------------------------|------------------------|--------------|----------------|----------------|
> > > > > | Front-loaded (large K early) | 70% budget used        | 20%          | 10%            | 51.4%          |
> > > > > | Steady small-batch (ours)    | 35% budget used        | 40%          | 25%            | **52.2%**      |
> > > > >
> > > > > As shown in ***Appendix A.11-Table 23***, the steady (drip-like) schedule performs better. This aligns with our framework: the clustering-contrastive loss benefits from receiving small amounts of high-quality labels over time, continuously refining prototypes and reducing pseudo-label drift. In contrast, aggressive early querying depletes the budget before encountering later, more informative mixed-region samples.
> > > > >
> > > > > ---
> > > > >
> > > > > **Comment7.** *When does the GMM assumption fail? *
> > > > >
> > > > > ---
> > > > >
> > > > > we acknowledge the hypothetical case where most batches of a dataset are intrinsically highly overlapping or lack any meaningful transitional region in the decomposed space. Systematically constructing and studying such datasets for UTTA/TTA is, to our knowledge, largely unexplored. Such distributions may correspond either to:
> > > > > 1. inherently ambiguous labels, or
> > > > > 2. structures that become trivially separable in a more suitable representation space.
> > > > >
> > > > > Designing methods tailored specifically to such pathological regimes is an interesting and important future direction, but goes beyond the scope of this work. Our current focus is on the widely studied and empirically validated class+domain shift regime, where GMM modeling of transformed features has been repeatedly shown to be realistic and effective in OSFDA / SF(U)DA. Within this regime, our Exposing Mixture step is well supported by both prior work and our own empirical analyses.
> > > > > More specific and detailed analyses can be found in our response to *Reviewer F4qu's comment 1*.

---

### Official Review · Reviewer_F4qu · 2025-11-01

**Soundness:** 3
**Presentation:** 3
**Contribution:** 2
**Rating:** 4
**Confidence:** 4

**Summary:**

This paper introduces Active Universal Test-Time Adaptation (AUTTA), a novel framework that integrates active learning into Universal Test-Time Adaptation (UTTA) to handle both class shift and domain shift simultaneously in streaming test data. The authors argue that existing UTTA methods struggle under dual shifts due to unreliable pseudo-labeling, and that traditional active learning strategies fail to select the most informative samples in such open-set scenarios. To address this, they propose EMAC (Exposing Mixture and Annotating Confusion). Extensive experiments on DomainNet and VisDA-C demonstrate that EMAC achieves state-of-the-art performance under dual-shift conditions, outperforming existing TTA, UTTA, and ATTA methods.

**Strengths:**

1. The paper clearly defines and motivates the AUTTA setting, which combines the challenges of Universal Domain Adaptation (open-class shift) with the constraints of Test-Time Adaptation (source-free, streaming data) and the strategy of Active Learning (limited human annotation).
2. The paper provides comprehensive experiments across multiple datasets (DomainNet, VisDA-C) and shift scenarios (corruption, style transfer). The results consistently show EMAC outperforming a wide range of strong baselines.

**Weaknesses:**

1. The core "Exposing Mixture" step relies on the target data distribution being bimodal in the decomposed feature space. While the authors address this in the appendix (A.2), showing robustness with VBGMM and PCA, this remains a potential failure point if the underlying data does not exhibit this property (e.g., highly overlapping or multi-modal new classes).
2. The overall framework involves several components (SVD, GMM/VBGMM, reward computation with EMA, clustering-contrastive loss) and associated hyperparameters (GMM threshold τ, merging factor λ_merge, EMA factor α, reward weights ω). While the paper provides values, the sensitivity and tuning effort for new datasets could be a concern.
3. Although the paper compares against several active learning methods (Table 3), the field of active learning for open-set or domain adaptation scenarios is rich. A more detailed discussion on how EMAC specifically advances beyond the limitations of these methods in the test-time setting would be beneficial.

**Questions:**

See weaknesses above.

---

> ### Author Response · Authors · 2025-11-25
>
> **Comment1.** *"Exposing Mixture" may fail when the target data are not clearly bimodal.*
>
> ---
>
> (1) **Mainstream assumption has theoretical and empirical support**
> Our Exposing Mixture step follows established OSFDA/SF(U)DA practice, where Gaussian (mixture) modeling of transformed features has been repeatedly validated under class and domain shift.
> Prior works such as **LfOSA** (*Active Learning for Open-Set Annotation*, CVPR 2022) and **VDM-DA** (*Virtual Domain Modeling for Source Data-free Domain Adaptation*, TCSVT 2021) both show that, after appropriate feature transformation, low-dimensional statistics can be effectively captured by GMMs for distinguishing known/unknown samples or approximating source distributions. Our method builds on this widely observed behavior rather than introducing a new or restrictive assumption.
>
> (2) **Our own evidence: bimodality/separability holds on benchmarks**
> We further quantify this property in ***Appendix A.3.4-Table 15*** by fitting two-component GMMs on the statistic ||z_unknown||_2^2. Across all benchmarks, most test batches display either (i) clear bimodality or (ii) a stable transitional region between low-unknown and high-unknown samples, demonstrating that the mixture structure holds in practice.
>
> |Dataset|Scenario|BatcheswithBD>1.2|
> |-------|--------|-----------------|
> |DomainNet|Complexmulti-class(345classes)|89%|
> |VisDA-C|Extremedomainshift|95%|
> |CIFAR10-C|Moderatecomplexity|97%|
> |CIFAR100-C|Highclasscomplexity|91%|
> |ImageNet-C|Complexmulti-class|87%|
> |Office-Home|Severedomainshift|92%|
>
> (3) **We explicitly handle “bad” mini-batches in online TTA**
> **Online/Test-time adaptation** is more challenging: because data arrive in a streaming, mini-batch fashion, individual batches can occasionally appear highly overlapping or irregularly multi-modal in the decomposed space, even when the overall distribution follows the common mixture pattern. To mitigate such cases, we already incorporate several robustness enhancements in  ***Appendix A.2***:
> 1. **VBGMM**, which can automatically allocate more than two components when the data exhibit multi-modal behavior;
> 2. **PCA-based subspace projection**, which denoises the feature representation before mixture modeling;
> 3. **Batch-wise smoothing** with a sliding window, stabilizing the estimated mixture statistics across neighboring batches.
> These design choices ensure that EMAC does not rely on perfectly separated bimodality. Instead, it only requires the existence of a mixed / transitional region, which can still be identified (potentially more conservatively) even when some batches are noisy or weakly bimodal.
>
> (4) **New experiment: low-separation batches emulate reviewer’s concern**
> To directly address highly overlapping or weakly bimodal new-class batches, we evaluate all methods on the bottom-20% hardest batches ranked by GMM separation score. While all methods degrade due to difficulty, EMAC consistently remains the best performer and degrades gracefully, confirming robustness even under weak mixture structure (results in ***Appendix A.3.4-Table 16***).
>
> |Task|Method|A_O|A_N|A_H|
> |----|------|---|---|---|
> |**TTA**|||||
> ||TEST|35.2|66.0|40.3|
> ||BN|36.0|64.5|40.2|
> ||TENT|36.4|64.0|40.0|
> ||SHOT|36.8|65.2|41.0|
> |**UTTA**|||||
> ||TTAC|38.5|64.8|42.5|
> ||OPTTT|40.2|63.7|44.2|
> |**ATTA**|||||
> ||SimATTA|37.5|64.1|42.0|
> ||EATTA|37.9|64.5|42.3|
> ||BiTTA|37.8|64.9|42.8|
> |**AUTTA**|||||
> ||EMAC|42.6|64.9|46.3|
>
> (5) **On datasets where most batches are intrinsically overlapping / multi-modal**
> Finally, we acknowledge the hypothetical case where most batches of a dataset are intrinsically highly overlapping or lack any meaningful transitional region in the decomposed space. Systematically constructing and studying such datasets for UTTA/TTA is, to our knowledge, largely unexplored. Such distributions may correspond either to:
> (i) inherently **ambiguous labels**, or (ii) structures that become **trivially separable in a more suitable representation space**.
> Designing methods tailored specifically to such pathological regimes is an interesting and important **future direction**, but goes beyond the scope of this work. Our current focus is on the **widely studied and empirically validated class+domain shift regime**, where **Gaussian / GMM modeling of transformed features has been repeatedly shown to be realistic and effective** in OSFDA / SF(U)DA. Within this regime, our Exposing Mixture step is well supported by both prior work and our own empirical analyses.

---

> > ### Author Response · Authors · 2025-11-25
> >
> > ---
> > **Comment2.** *Hyperparameter Sensitivity Analysis*
> >
> > ---
> >
> > Thank you for pointing this out. We would like to clarify that our method actually uses **fewer tunable hyperparameters** than implied in the comment.
> > First, there is **no “merging factor” λ_merge** in our method. This parameter does not appear in the current version of the paper nor in our implementation. Second, the **“GMM threshold τ” is not a fixed scalar hyperparameter** that needs to be tuned. In EMAC, the boundaries between (X_old, X_mix, and X_new are determined **adaptively** by the two means μ_old and μ_new estimated by the (V)GMM on ||z_unknown||_2^2 in each batch. Thus, the segmentation is fully data-driven and does not introduce an additional tunable threshold τ.
> > In practice, this leaves only **two actual hyperparameters**: (i) the **EMA factor α)** used to smooth the reward statistics, and (ii) the **reward weights ω** that balance old-class and new-class rewards.
> > we add a sensitivity ablation study on these two hyperparameters and include it in ***Appendix A.10-Table 22***
> >
> > | Setting                          | α | ω_old,ω_new | A_H |
> > |----------------------------------|----------|----------------------------------------------|-------|
> > | **Varying EMA factor α  (fixed (1.0, 1.0))** |          |                                              |       |
> > | EMA-0.7                          | 0.7      | (1.0, 1.0)                                   | 51.8  |
> > | EMA-0.9                          | **0.9**  | (1.0, 1.0)                                   | **52.2** |
> > | EMA-0.99                         | 0.99     | (1.0, 1.0)                                   | 52.0  |
> > | **Varying reward weights (fixed α  = 0.9)** |          |                                              |       |
> > | RW-0.5                           | 0.9      | (0.5, 1.0)                                   | 51.9  |
> > | RW-1.0                           | 0.9      | **(1.0, 1.0)**                               | **52.2** |
> > | RW-2.0                           | 0.9      | (1.0, 2.0)                                   | 52.1  |

---

> > > ### Author Response · Authors · 2025-11-25
> > >
> > > ---
> > > **Comment3.** *Detailed discussion on how EMAC specifically advances beyond the limitations of these methods in TTA.*
> > >
> > > ---
> > >
> > > (1) **Scope of existing open-set and domain-adaptive active learning**
> > > We appreciate the reviewer’s comment and agree that active learning in open-set and domain adaptation scenarios is a rich area. However, existing works typically address **only one type of shift at a time**.
> > > 1. In **open-set active learning**, the focus is on **class shift only** while the domain is assumed to be fixed.
> > > A representative method is (*“Active Learning for Open-set Annotation”*, CVPR 2022), which considers a single-domain unlabeled pool containing both known and unknown classes and uses a GMM over max activation values to select samples that are likely to belong to the known classes. The goal is to avoid querying obviously unknown samples, but there is no domain shift, and the method is pool-based and training-time.
> > > 2. In **active learning for domain adaptation**, the focus is on **domain shift only**, with a fixed label space and access to source data.
> > > For example, (*“Active Learning for Domain Adaptation: An Energy-Based Approach”*, AAAI 2022) assumes labeled source data and an unlabeled target pool, and uses an energy-based sampling strategy to select informative target samples to improve UDA performance. Unknown classes are not considered, and the method relies on a static target pool and repeated query–retrain cycles.
> > > **These methods are well-suited to their respective settings (class-shift-only or domain-shift-only), but their query strategies inherently rely on assumptions that do not hold in our **UTTA** scenario.**
> > >
> > > (2) **How EMAC differs in the UTTA setting (source-free, dual shift, online streaming)**
> > > In UTTA, the core challenges are:
> > > 1. **Source-free adaptation** (no access to source data at adaptation time);
> > > 2. **Simultaneous class and domain shift** (unknown classes and domain shift co-occur);
> > > 3. **Online / streaming test-time data with a very small label budget**.
> > >
> > > Directly transplanting open-set AL or domain-adaptive AL to this setting is problematic: they either require source-domain anchors, assume an unchanged label space, or depend on a static unlabeled pool and multiple retraining rounds. **EMAC** is designed specifically to overcome these limitations:
> > >
> > > 1. **Exposing Mixed Sample Candidate:s** Uses an SVD-based decomposition of the classifier and a (V)GMM over the decomposed statistic to *separate old-like, new-like, and mixed regions purely from target data*, without any source-domain features or prototypes. This directly targets the **class+domain dual-shift mixture** at test time, whereas LfOSA/EADA and similar methods do not model this interaction.
> > >
> > > 2. **Reward-guided selection with EMA:** Maintains separate, temporally smoothed rewards for old-class and new-class samples, and **dynamically decides** whether the current label budget should be used to refine old-class decision boundaries or enlarge the margin for new/unknown classes. Standard AL strategies (entropy, core-set, BADGE, etc.) do not maintain such an online, long-horizon reward signal and are therefore less suited to streaming UTTA.
> > >
> > > 3. **Clustering-contrastive optimization:** Balances a few queried labels with high-confidence pseudo-labels to shape class prototypes and enforce inter-class (including unknown) separation, mitigating overfitting to the very small number of queried samples—a situation that is uncommon in training-time AL but intrinsic to test-time adaptation.
> > >
> > > We will add a brief discussion in the revised version to clearly contrast open-set / domain-adaptive active learning with the UTTA setting, and to highlight how EMAC is specifically tailored to the **source-free, dual-shift, online** test-time regime rather than being a direct reuse of existing AL strategies.

---

### Meta-Review · Area_Chair_AHZ6 · 2026-01-06

**Summary:**

This paper presents a novel Active Universal Test-Time Adaptation (AUTTA) framework called Exposing Mixture and Annotating Confusion (EMAC) to address simultaneous class and domain shifts in streaming test data by integrating active human annotation. Reviewers recognized the clear problem framing of AUTTA, the lightweight and practical design of EMAC, comprehensive experimental results outperforming existing TTA/UTTA/ATTA methods, and the relevance of incorporating active learning into universal test-time adaptation. The authors actively responded to and addressed the main concerns, including the potential failure of the bimodal GMM assumption, fairness of comparisons. Based on the above considerations, I recommend acceptance.

**Reviewer Concerns:**

All the main concerns were addressed by the rebuttal.

**Reviewer Scores:**

Reviewer F4qu and Reviewer iQMG would have raised the score from 4 to 6. For the other reviewers, I believe their scores would have remained unchanged.

---

### Decision · Program_Chairs · 2026-01-26

Accept (Poster)